# Evolution of the stratospheric polar vortex edge intensity and duration in the Southern hemisphere over the 1979 – 2020 period

Audrey Lecouffe[1], Sophie Godin-Beekmann[1], Andrea Pazmiño[1], and Alain Hauchecorne[1]

[1]LATMOS/IPSL, UVSQ, Sorbonne Université, CNRS, Paris, France

**Correspondence:** Audrey Lecouffe (audrey.lecouffe@latmos.ipsl.fr)

**Abstract.** The intensity and position of the Southern Hemisphere stratospheric polar vortex edge is evaluated as a function of equivalent latitude over the 1979 – 2020 period on three isentropic levels (475 K, 550 K and 675 K) from ECMWF ERA-Interim reanalysis. The study also includes an analysis of the onset and breakup dates of the polar vortex, which are determined from wind thresholds (e.g. 15.2 m.s$^{-1}$, 20 m.s$^{-1}$ and 25 m.s$^{-1}$) along the vortex edge. The vortex edge is stronger in late
winter, over September - October - November with the period of strongest intensity occurring later at the lowermost level. At the same period, we observe a lower variability of the edge position. Long-term increase of the vortex edge intensity and break-up date is observed over the 1979 – 1999 period, linked to the increase of the ozone hole. Long-term decrease of the vortex onset date related to the 25 m.s$^{-1}$ wind threshold is also observed at 475 K during this period. The solar cycle and to a lower extent the quasi-biennal oscillation (QBO) and El Niño Southern Oscillation (ENSO) modulate the inter-annual
evolution of the strength of the vortex edge and the vortex breakup dates. Stronger vortex edge and longer vortex duration is observed in solar minimum (minSC) years, with the QBO and ENSO further modulating the solar cycle influence, especially at 475 K and 550 K: during West QBO (wQBO) phases, the difference between vortex edge intensity for minSC and maxSC years is smaller than during East QBO (eQBO) phases. The polar vortex edge is stronger and lasts longer for maxSC/wQBO years than for maxSC/eQBO years. ENSO has a weaker impact but the vortex edge is somewhat stronger during cold ENSO
phases for both minSC and maxSC years.

## 1 Introduction

The stratospheric polar vortex is a seasonal low-pressure system characterized by a strong wind belt that isolates polar air from lower latitudes. It appears due to the seasonal cooling associated with the decrease of solar radiation above the pole (Randel and Newman, 1998). As the incident solar energy decreases, and the gradient of temperature between the pole and the tropics
becomes stronger, the strength of the stratospheric westerly winds increases. When the winds reach a critical value, a large-scale vortex is formed, which extends from the lowermost stratosphere to the stratopause. Depending on altitude, the maximum area encompassed by the polar vortex exceeds millions of square kilometers (NOAA, 2021). Above an altitude of about 14 km, the vortex edge region is stable and constitutes a powerful barrier, preventing mixing of cold polar air with warmer air masses from lower latitudes. Over Antarctica, the polar vortex is generally present from April until December with a large variability in the
breakup dates resulting from the year-to-year variability of dynamical processes in the stratosphere (Waugh and Randel 1999;

Rao and Garfinkel 2021). Conversely, the less stable Arctic polar vortex has more year-to-year variability (e.g. Andrews et al. 1987; Hu et al. 2014; Butler et al. 2019). It forms in November and lasts until the end of February or early April, depending on the year. The stratospheric polar vortex have been the subject of studies linked to the ozone layer depletion, which started in the late 1970 (Farman et al., 1985; Solomon 1999). Ozone loss occurs in both hemispheres. This loss is variable in the northern

hemisphere as many studies have shown (Solomon 1999; Goutail et al. 2005; Pommereau et al. 2018; WMO 2018; Grooß and Müller 2020). In the Southern Hemisphere, the ozone hole, defined as an area with total ozone values less than 220 DU, has become a recurring seasonal phenomenon. Ozone destruction begins in late winter, close to the edge region of the polar vortex, as solar radiation increases over the Pole. The destruction of ozone inside the southern vortex accelerates from late August until late September or early October, reaching an almost complete destruction of ozone in the lower stratosphere. The depletion

of the ozone layer is caused by anthropogenic emission of ozone depleting substances (ODS - mainly chlorofluorocarbons and halons and their industrial substitutes), which enhances ozone destruction cycles by halogen compounds. This depletion is largest in the polar vortex due to the activation of chlorine species through heterogeneous reactions that take place at the surface of polar stratospheric clouds (PSCs) which form in the cold polar vortex (Solomon, 1999). The increase in solar radiation over the Pole at the end of winter triggers rapid chemical cycles which quickly destroy ozone, leading to the appearance of the

well-known ozone hole over Antarctica (e.g. WMO, 2018). By the end of spring, stratospheric temperatures increases, the polar vortex breaks up and ozone depleted air masses dilute into the Southern Hemisphere. From one year to the next, the severity of the ozone hole depends on the strength of the polar vortex, its minimum temperatures and duration. The future recovery of the ozone layer and disappearance of the ozone hole depends on the polar vortex evolution under the influence of both the decrease of ODS abundance in the stratosphere and the increase of greenhouse gases (GHG) as both phenomena

impact radiative, dynamical and chemical processes in the stratosphere. Many studies document this phenomenon (e.g. WMO 2018 and references therein). The polar vortex also has an impact on the climate surface in both hemispheres. Indeed, studies have shown an effect of the stratospheric polar vortex displacements on cold spells in the northern hemisphere, in North America (Tripathi et al., 2015). In the southern hemisphere, others have shown that a weak vortex can have an influence on the surface climate in Australia. Lim et al. (2019) have highlighted that selected years of lower vortex intensity results in

higher temperatures and less precipitation over eastern Australia. The dramatic weakening of the Antarctic vortex in 2019 had a large impact on meteorological conditions over the country that resulted in the strong Australian fires of the turn of the year 2019/2020.

The inner vortex is characterized by high absolute values of potential vorticity (PV). As this parameter is conserved on isentropic surfaces during weeks, PV maps on such surfaces is one of the primary diagnostic tools for the dynamical processes

analyses in the stratosphere and inside the polar vortex. McIntyre and Palmer (1983) first represented daily PV global maps of isentropic surfaces, demonstrating a material separation in the stratosphere between the main vortex, characterized by high absolute PV values, the surf zone by weak absolute PV values, and a zone of strong meridional PV gradient in between: the so-called vortex boundary or vortex edge, which is an area of low mixing representing a dynamical barrier to air masses exchanges. Numerous studies on the vortex boundary definition have been performed. Nash et al. (1996) defined the vortex edge as the

location of the maximum PV gradient as a function of equivalent latitude (EL), weighted by the mean wind speed. EL defines

the latitude limit of the polar area which exceeds a certain PV value (maximum PV is then given at EL = 90 degrees, e.g. Butchart and Remsberg (1986)). The mean wind speed is the mean of the wind values around an equivalent latitude contour. A PV field sorted by EL will then make the polar vortex concentric around the pole. This is the method used in this study. Nakamura (1996) has developed the effective diffusivity diagnostic, which is applied on tracers to identify transport barriers and mixing regions. Hauchecorne et al. (2002) used this method to quantify the transport of polar vortex air to mid-latitudes, as well as to evaluate the polar vortex barrier intensity. The method of elliptical diagnostics of a contour used by Waugh (1997), consists in fitting an ellipse to the contour of a parameter. It subsequently determines several variables of this ellipse, for example latitude and longitude of the center, the equivalent latitude, or its orientation. It is possible to calculate the elliptical diagnostics of a contour of conservative tracers such as PV or long-lived chemical species around the polar vortex edge region (Waugh and Randel, 1999). The vortex forms in autumn, intensifies throughout the winter and disappears in spring/summer. Its overall strength is variable from one year to the next. Different studies have analyzed the inter-annual variability of the polar vortex induced by forcings such as the solar flux (SF), Quasi-Biennial Oscillation (QBO) and El Niño Southern Oscillation (ENSO), particularly in the Northern Hemisphere. QBO is a quasi-periodic oscillation of the equatorial zonal wind between easterlies and westerlies. Holton and Tan (1980) made a composite study of zonal wind in the Northern Hemisphere at 50 hPa from 1962 to 1977 based on the different QBO phases. They showed that the vortex is less disturbed during the West phase of the QBO (wQBO) at 50 hPa than during the East phase (eQBO). Labitzke and Van Loon (1988) evaluated the temperature and strength of the Arctic polar vortex according to the solar cycle and the QBO. They found that the vortex is warm and weak during solar maxima/eQBO phases, and cold and strong during solar minima/wQBO phases at 50 hPa. Camp and Tung (2007) supports this finding that the state of the northern hemisphere polar stratosphere is less perturbed during solar cycle minimum and westerly QBO phases. Then, Baldwin and Dunkerton (1998) showed over a period of 18 years, that the Antarctic polar vortex at 10 hPa is slightly colder during wQBO. ENSO is an irregular oscillation in winds and sea surface temperatures over the tropical eastern Pacific Ocean, affecting the climate of the tropics and subtropics. It influences also other climatic parameters such as precipitations worldwide and ozone levels in the lower stratosphere (WMO, 2018). Domeisen et al. (2019) have indicated that the El Niño events are associated with a warming and weakening of the polar vortex in the polar stratosphere in both hemispheres, and Li et al. (2016) have shown that early breakup of the southern polar vortex occurs during El Niño events. In contrast, Rao and Ren (2020) did not find a significant impact of the canonical ENSO index on the Southern Hemisphere polar vortex in both observations and modeling studies. With indices of Niño-3 and Niño-4 regions, Hurwitz et al. (2011) have shown that during "warm pool event" (positive SST in Niño-4 regions) the heat flux is higher and the Antarctic vortex breaks up earlier. Several methods have been suggested in order to determine the onset and breakup dates of the polar vortex. They are based on a minimum area computed from equivalent latitudes (Manney et al. 1994; Zhou et al. 2000) or mean wind speed thresholds along the edge (e.g. Nash et al. 1996). The latter is used in WMO (2018) to calculate the dates at which the Arctic and Antarctic polar vortex breaks each spring.

The objective of this paper is to analyze the long-term evolution of the intensity, position and duration of the Southern polar vortex edge as a function of equivalent latitude over several decades (1979 – 2020). ERA-Interim reanalyses and operational data from the European Centre for Medium-Range Weather Forecast (ECMWF) are used for the study, which includes an

evaluation of the onset and breakup dates of the polar vortex over the period. At an inter-annual scale, the signature of the 11-year solar cycle, QBO and ENSO, is evaluated on the vortex edge evolution. This is the first study of the variability of the Antarctic stratospheric polar vortex edge and persistence over a long period (42 years).

The paper is organized as follows. Section 2 presents the ECMWF dataset and the data sources of the forcings (SF, QBO,
and ENSO) used for the analysis of inter-annual variability of the polar vortex edge. Section 3 describes the methods used in the study, such as the MIMOSA (Modélisation Isentrope du transport Méso-échelle de l'Ozone Stratosphérique par Advection) model (Hauchecorne et al., 2002), which is used to construct the PV maps as a function of potential temperature and equivalent latitude. The methods used for the vortex edge characterization and for determining the onset and breakup dates of the polar vortex are also discussed in this section. Section 4 presents the statistical analysis of the annual evolution of the vortex edge
over the studied period as well as its inter-annual evolution, related to the SF, QBO and ENSO forcings, while results on the inter-annual evolution of the vortex onset and breakup dates are given in Section 5. Further discussion of the results and perspective of the study are presented in Section 6.

## 2   Data

### 2.1   Potential vorticity fields

PV fields are calculated from ECMWF ERA-Interim reanalysis [1] (Dee et al., 2011). As these reanalyses end in August 2019, we used the operational data from ECMWF from September 2019 until December 2020. Recently, Millan et al. (2020) compared the polar vortex evolution with different reanalyses, including ERA-Interim. Results showed that all reanalyses where in agreement with the reanalysis ensemble mean (REM), which shows that we can be confident with the ERA-Interim reanalyses for our study. ERA-Interim temperature, geopotential and wind data with a resolution of 1.125° latitude x 1.125°
longitude are inputs to the MIMOSA model, which is a three-dimensional high-resolution PV advection model (Hauchecorne et al., 2002). From MIMOSA high resolution PV fields it is possible to follow the evolution of polar air masses and filamentation processes of the polar vortex. Sampled every 6 hours, ERA-Interim reanalyses are interpolated on selected isentropic surfaces. The model computes PV and EL fields on the isentropic surfaces with a resolution of 0.3° latitude x 0.3° longitude, using a polar projection centered on the South from 90°S to 10°N. The advection method is applied to this orthographic grid. After
some time, the MIMOSA grid is distorted by the horizontal gradients of the wind fields. A re-interpolation of the PV fields on the original grid every 6 hours is then performed. Finally, in order to take into account diabatic processes, a relaxation of the MIMOSA advected PV (APV) towards the ECMWF PV is made every 12 hours with a 10 day time constant. This model has been used to analyze, among other studies, the permeability of the southern polar vortex to volcanic aerosols from Cerro Hudson and Mount Pinatubo eruptions in 1991 (Godin et al., 2001), and to predict the extension in the lower mid-latitude
stratosphere of polar and subtropical air masses (Heese et al., 2001). In Pazmino et al. (2018), PV fields simulated by the model are used to evaluate average total ozone evolution within the Antarctic vortex. For this study, PV fields are computed at 675 K, 550 K and 475 K isentropic levels.

## 2.2 Forcings of interannual variability

Forcings considered for the analyses of the inter-annual variability of the vortex edge are described in Table 1. For the solar flux, we are mainly interested in the variability induced by the 11-year solar cycle. The F10.7 solar flux data covers six solar cycles, including those covering our study period, the last four. It correlates well with the 11-year sunspot cycle (Mishra et al., 2005, Tiwari and Kumar, 2018) and has been used frequently as a proxy for solar activity (e.g. Solomon, 1999; Gray, 2003; Pazmino et al., 2018). It is defined in solar flux units (1 sfu = $10^{-22}$ W.m$^{-2}$.Hz$^{-1}$). For our study, we averaged the 10.7 cm solar flux and other proxies over the May - November period, which corresponds to the time period when the Southern polar vortex is well formed. Data were obtained for solar cycles 21 to 24 (1976 to 2020). Years characterized by minimum and maximum solar intensity were selected from the difference of maximum and minimum intensity of each cycle (a methodology also considered in Rao et al. 2019). The minimum (maximum) intensity threshold was defined as the lower (upper) third of this difference, so that the minimum and maximum thresholds are different for each cycle. The selection results in 15 maximum solar (maxSC) years and 20 minimum solar (minSC) years over the whole study period. In order to investigate the influence of QBO on the polar vortex, we used Singapore monthly mean zonal wind at the 50 hPa level, and averaged this parameter each year during the same period as for the solar cycle. QBO is sorted by negative phase for East QBO (eQBO) with 19 years and positive phase for West QBO (wQBO) with 23 years. In the case of El Niño Southern Oscillation, the Multivariate ENSO Index (MEI) version 2 was used in this study. It correspond to the combination of empirical orthogonal function (EOF) of sea level pressure (SLP), sea surface temperature (SST), zonal and meridional components of surface wind and outgoing longwave radiation in the tropical Pacific basin. Referring to the NOAA description of the MEI.v2 index (see data availability [4]): "The EOF are calculated for 12 overlapping bi-monthly "seasons" in order to take into account ENSO's seasonality, and reduce effects of higher frequency intra-seasonal variability". Then mean ENSO over the period is sorted to distinguish La Niña, characterized by negative values smaller than -0.5 MEI.v2 (cold ENSO), and El Niño by positive values higher than +0.5 MEI.v2 (warm ENSO). Then 10 wENSO and 14 cENSO years are considered in this study.

**Table 1.** Proxies: source, characteristics and period.

| Proxy | Source | Characteristics | Period |
|-------|--------|-----------------|--------|
| SF | Dominion Radio Astrophysical Observatory (National Research Council Canada) [2] | Monthly mean solar flux at 10.7 cm | May - November |
| QBO | Institute of Meteorology (Freie Universität Berlin) [3] | Monthly mean quasi-biennial oscillation at 50 hPa | May - November |
| ENSO | NOAA Earth System Research Laboratory [4] | Bi-monthly Multivariate ENSO index (MEI.v2) | May - November |

## 3 Methods

### 3.1 Vortex edge characterization

As mentioned in the introduction, the vortex edge is characterized by a strong PV gradient. To represent the vortex edge position, the method described in Nash et al. (1996) is used, which consists in determining the position of the edge from the maximum PV gradient weighted by the mean wind speed as a function of EL. The maximum gradient is evaluated in the [-85°, -40°EL] range in order to avoid high PV values at the pole and disturbances by the subtropical jet. The position of the edge is defined by the EL of the $max(dPV/dEL \times W(EL)[-85°, -40°EL])$ where W is the mean wind speed.

### 3.2 Determination of polar vortex onset and breakup dates

Several methods have been used in order to determine the onset and breakup dates of the polar vortex in the Northern hemisphere (NH), as mentioned previously. Manney et al. (1994) first determined that the breakup date corresponds to the date when the EL of a chosen PV contour at the 465 K level is greater than 80°, using PV data computed from the National Centers for Environmental Prediction and the National Center for Atmospheric Research (NCEP/NCAR) reanalyses. From a given PV contour, the authors determined that if the corresponding EL position is poleward of 80°LE, then the vortex is not well formed. This defines the duration of the polar vortex. Subsequently, using wind fields in addition to the PV gradient as a function of EL, Nash et al. (1996) considered that the vortex is well formed at 450 K when the mean wind speed along the vortex edge is equal to or greater than 15.2 m.s$^{-1}$. They also used the 3.2 m.s$^{-1}$ standard deviation interval to provide a range of dates during which the vortex forms and breaks. Then Waugh et al. (1999) analyzed the breakup date of the Arctic and Antarctic polar vortex using NCEP data for the 1958-1999 period. They showed a tendency of extension of the breakup date after 1979 in the Antarctic that could be due to radiative processes induced by the lower ozone levels within the vortex. Zhou et al. (2000) used the same method as Manney et al. (1994) and compared the vortex breakup dates in the 1990s with those of the 1980s based on NCEP data, considering that the vortex breaks up and disappears when its size falls below 1% of the Earth's surface. The authors demonstrated that the Antarctic vortex lasted two weeks longer in the $1991 - 1998$ period than in the $1979 - 1984$ period. The authors joined other studies (Atkinson et al. 1989; Müller et al. 2008; Zhang et al. 2017) in concluding that the vortex lifetime is influenced by the ozone depletion during spring. Akiyoshi et al. (2009) used the same method as Nash et al. (1996) and added threshold values of 20 and 25 m.s$^{-1}$ to compare variations of breakup dates in model and observations over the $1980 - 2004$ period. In this study, we use the Nash et al. (1996) method to determine the vortex onset and breakup dates, also used in WMO (2018). Two threshold values (20 m.s$^{-1}$ and 25 m.s$^{-1}$) following Akiyoshi et al. (2009) are added to this method, in order to evaluate the sensitivity of the onset and breakup dates to the chosen threshold values (see section 5).

## 4  Evolution of the polar vortex edge throughout the winter

### 4.1  Intensity of the vortex edge

The statistical analysis of the evolution of the vortex edge intensity throughout the winter from 1979 to 2020 at the 675 K, 550 K and 475 K isentropic surfaces is shown in Figure 1, which displays the maximum PV gradient smoothed by 5 days running mean, in EL from May to December. In each panel, the black bold curve represents the median values and blue filled areas indicate values between $20^{th}$ and $80^{th}$ percentiles. Thin dark lines are the overall maximum and minimum over the 1979 – 2020 period. Data are considered every year between the onset and the breakup dates of the vortex (see section 5) and the percentiles, median and overall extrema are plotted for days with 3 years or more of data. The statistical parameters with at least 3 years of data are obtained until day 343, 354 and 361 at 675 K, 550 K and 475 K, respectively. Results show that the vortex is systematically present on May $1^{st}$, and reaches its maximum intensity during different periods of the winter depending on the level, e.g. later at the lower levels. It is reached from September to late October at 675 K with a median peak value of 20.8 PV units/°EL in October, from September to early November at 550 K with a peak value of 7.8 PV units/°EL at the beginning of October, and later for 475 K during the first half of November with a peak value of 3.9 PV units/°EL. This period of maximum intensity is also characterized by a larger variability (as seen from the maximum and minimum curves, especially for the lower isentropic levels). Depending on the year and the level, the vortex breaks up between mid-October and the end of December at the latest.

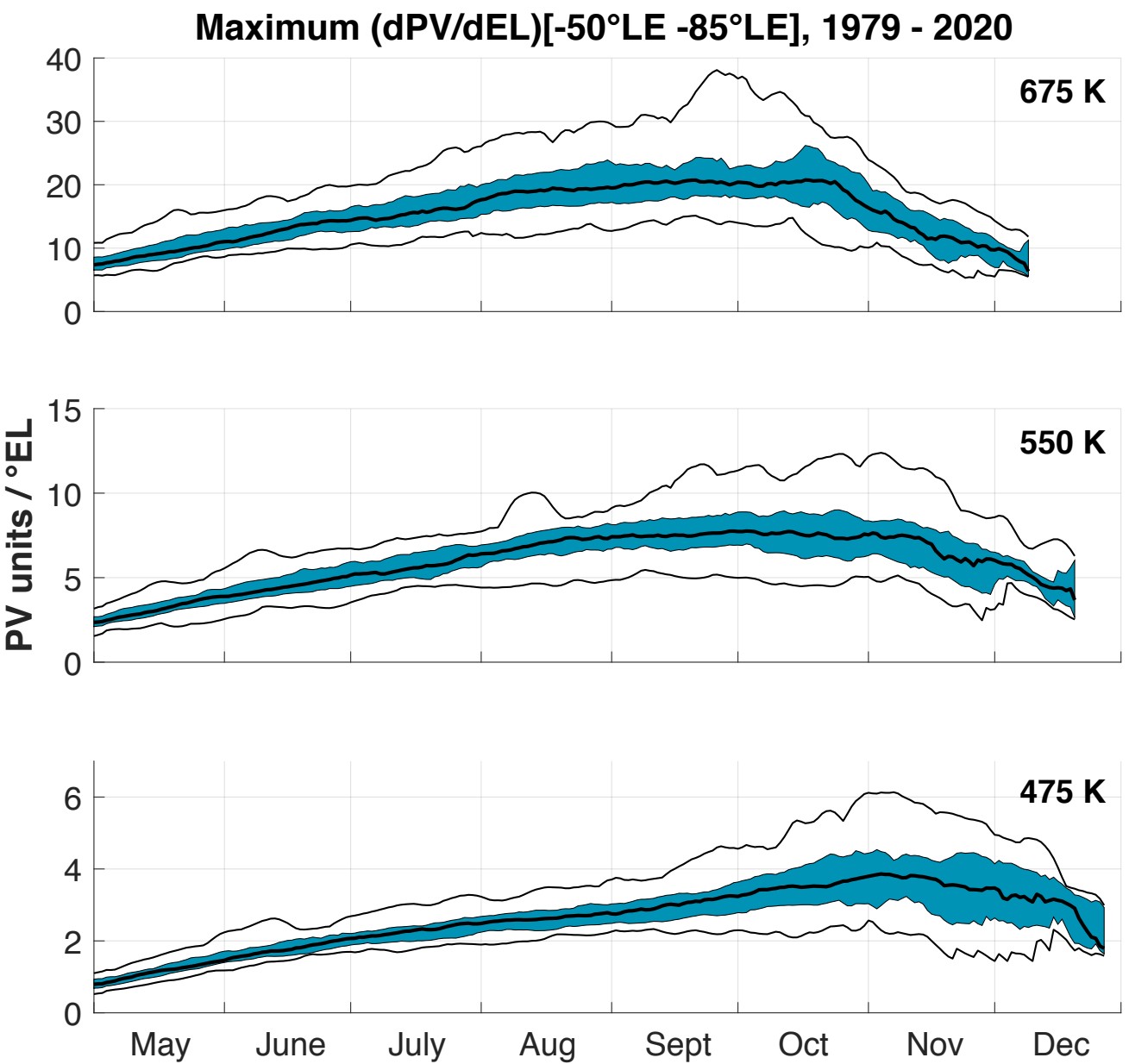

**Figure 1.** Evolution of daily maximum PV gradient in the 1979 – 2020 period, from top to bottom: 675 K, 550 K and 475 K. Median values are represented by the black bold curve. Blue filled area shows values between 20 and 80 percentiles, while thin black curves represent the maximum and minimum over the period.

Figure 2 represents the evolution of the vortex edge position in EL. For this parameter, medians and percentiles curves show a similar behavior for the various levels from May to late September for all levels. The polar vortex edge position is reached between mid-July and late August at 675 K, between mid-July and mid-August at 550 K and between mid-August and

September at 475 K, with respective median average values of -57.3°EL, -57.8°EL and -58.4°EL. The minima show clearly the large reduction in the vortex area due to the major warming in 2002 during October. It is less pronounced at 475 K where the edge position decreased to a minimum of -67.8°EL, compared to -76.3°EL and -71°EL at 675 K and 550 K, respectively (e.g. Hoppel et al. 2003). The 2019 winter impacts the minimum curve during the last 2 weeks of September at 675 K and is located

5  between the minimum curve and the 20th percentiles from September until the beginning of November for each levels. During this year, a minor SSW occurred at the end of August, which displaced and weakened the polar vortex. The stratospheric polar vortex abruptly weakened and warmed on August 25th (Lim et al., 2021). MERRA2 analyses showed a rapid 50 K increase of polar temperature at 10 hPa between September 5 and September 11 (Yamazaki et al., 2020). Minimum values of winds at 10 hPa and 60°S were found on September 18 (Rao et al., 2020). This event induced the smallest Antarctic ozone hole on

10  record. Although it appeared earlier than usual in August, the ozone hole reached an area of 15 million km$^2$ by September 1, but decreased to an area of 8 million km$^2$ by September 17 (Lim et al., 2021). The variability in vortex area decreases for all levels during the period of maximum edge intensity period: the EL difference between the 20 and 80 percentiles reduces to 3.7°EL in October at 675 K, and 3.1°EL at the 550 K and 475 K levels compared to 4.6°EL, 5.4°EL and 5.2°EL in August, respectively.

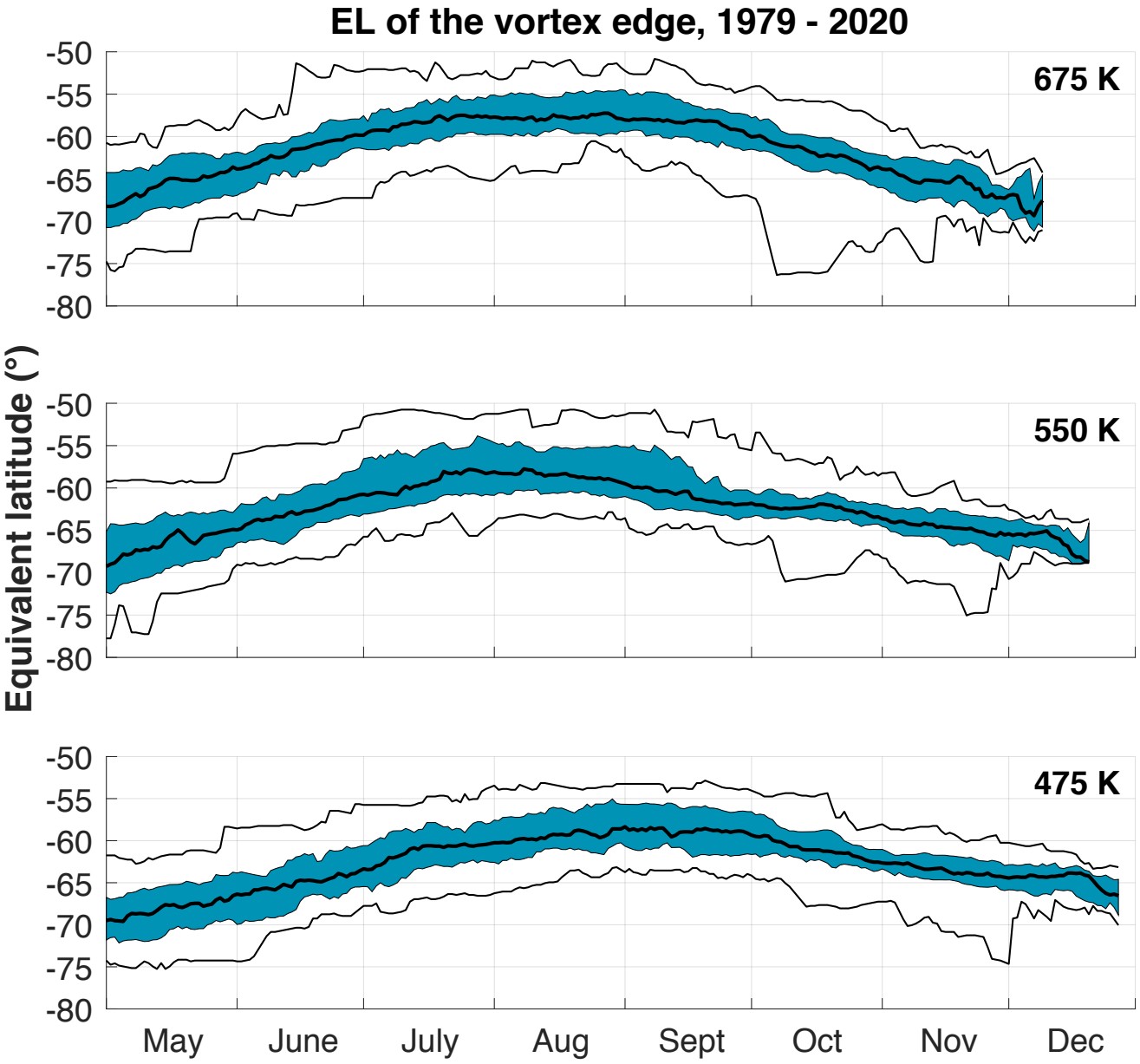

**Figure 2.** Evolution of daily position of the vortex edge in equivalent latitude as a function of time over the 1979 – 2020 period, from top to bottom: 675 K, 550 K and 475 K. Median values are represented by the black bold curve. Blue filled areas show values between 20 and 80 percentiles, while thin black curves represent the maximum and minimum values over the period.

## 4.2 Influence of Solar Cycle, Quasi-Biennial Oscillation and El Niño Southern Oscillation on the polar vortex edge

Factors such as the solar cycle, QBO and ENSO are used to describe the interannual variability in the temporal evolution of the polar vortex edge over the 1979 – 2020 period. As mentioned in the introduction, these variables were largely used in various studies of the stratospheric polar vortex.

### 4.2.1 The Solar Cycle

The intensity of the vortex edge has been sorted according to the maximum (maxSC) and minimum (minSC) solar activity years (see section 2). Figure 3 displays the composite analysis of the temporal evolution of the polar vortex edge intensity throughout the winter from 1979 to 2020 at the three isentropic levels. In each panel, the dark (light) grey area represents values between $20^{th}$ and $80^{th}$ percentiles of maxSC (minSC) years with the median in red (blue). The various panels of the figure show that minSC years are generally characterized by a stronger vortex edge. Also, maxSC years vortex break up earlier than during minSC, e.g. 6 days earlier at 675 K, 4 days at 550 K, and 3 days at 475 K. The relative difference between the maxSC and minSC medians in the periods of maximum intensity is larger at 550 K (16.4% relative difference) than at 475 K (13%), and 675 K (11.2%) levels. A Mann-Whitney test was performed to characterize the significance of these differences. The Mann-Whitney test results indicate that differences are significant from 27 September to 26 October at 675 K, from 9 to 24 September and from 3 October to 21 November at 550 K, and from 19 September to 15 October and from 11 to 26 November at 475 K. For the three levels, there is a jump in the vortex edge intensity for the maxSC years during November, which is not observed for minSC years. These jumps in the medians are related to smaller number of years included in the statistical parameters due to earlier vortex breakup dates for maxSC years.

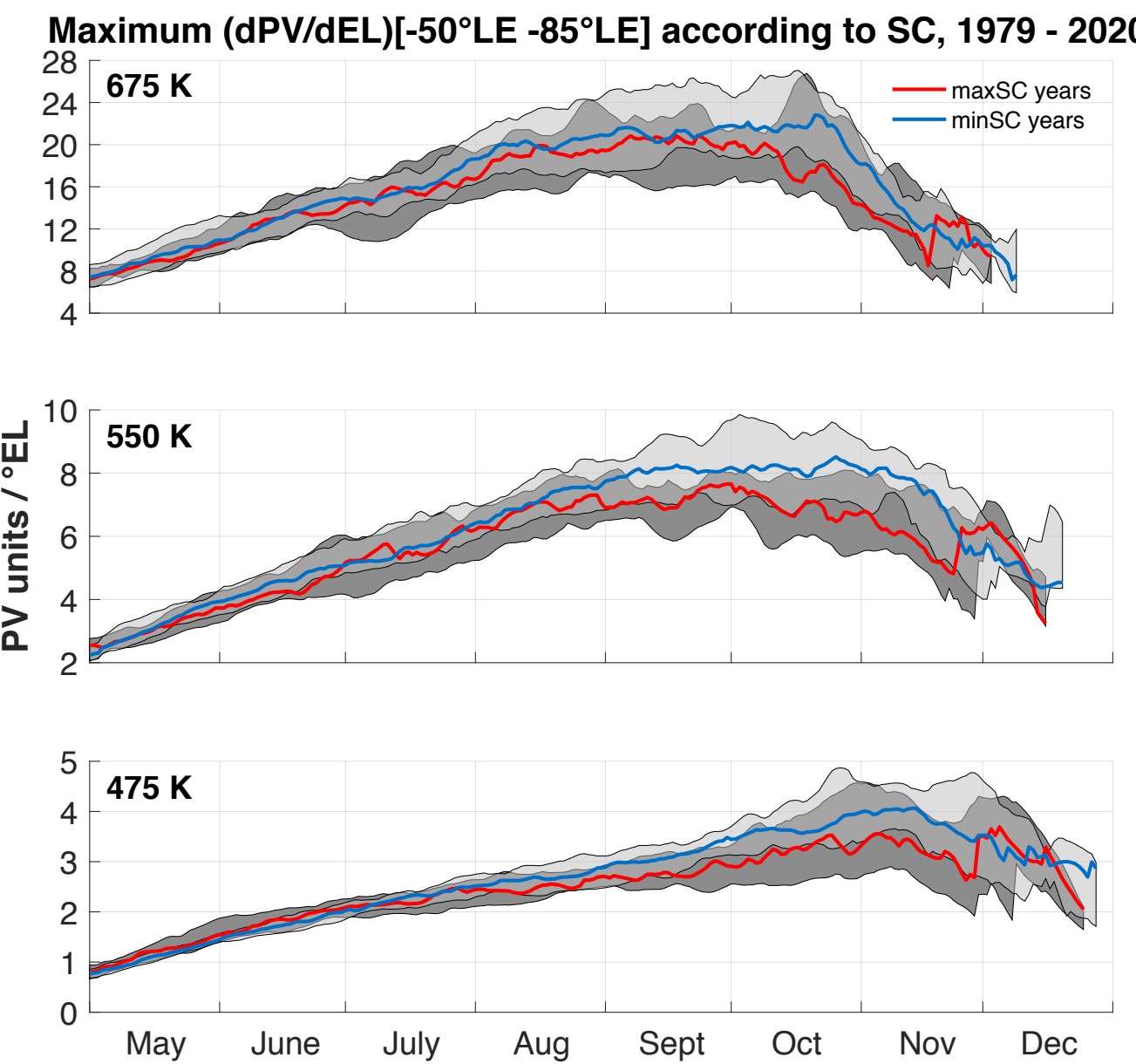

**Figure 3.** SC composites of vortex edge intensity's seasonal evolution for the 1979 – 2020 period, from top to bottom: 675 K, 550 K and 475 K. Red (blue) curves represent median values for maxSC (minSC) years. Dark (light) grey-filled areas indicate values between 20 and 80 percentiles for maxSC (minSC) years.

Figure 4 represents the composite analysis evolution of the vortex edge position according to SC in a similar way as in Figure 3 for the vortex edge intensity. Results do not show a large impact of the SC on the vortex edge position, although the vortex appears to be somewhat larger during maxSC periods, with also a larger variability. In the beginning of May, the vortex

edge extends to -68°EL, then reaches a maximum at -56.1°EL (-57.4°EL) during the maxSC (minSC) between late August and mid-September at 675 K. At 550 K and 475 K, maximum equivalent latitude positions reached according to the maxSC (minSC) years are -55.2°EL (-58.7°EL) between mid-July and August and -56.4°EL (-58.6°EL) between mid-August and September. There is less variability and fewer differences between maxSC and minSC years during the period of maximum intensity of the edge (see section 4.1). The difference between the medians was assessed by a Mann-Whitney and differences are significant from 9 to 18 September at 675 K, from 18 July to 11 August and from 27 August to 7 September at 550 K, and from 15 to 20 June at 475 K.

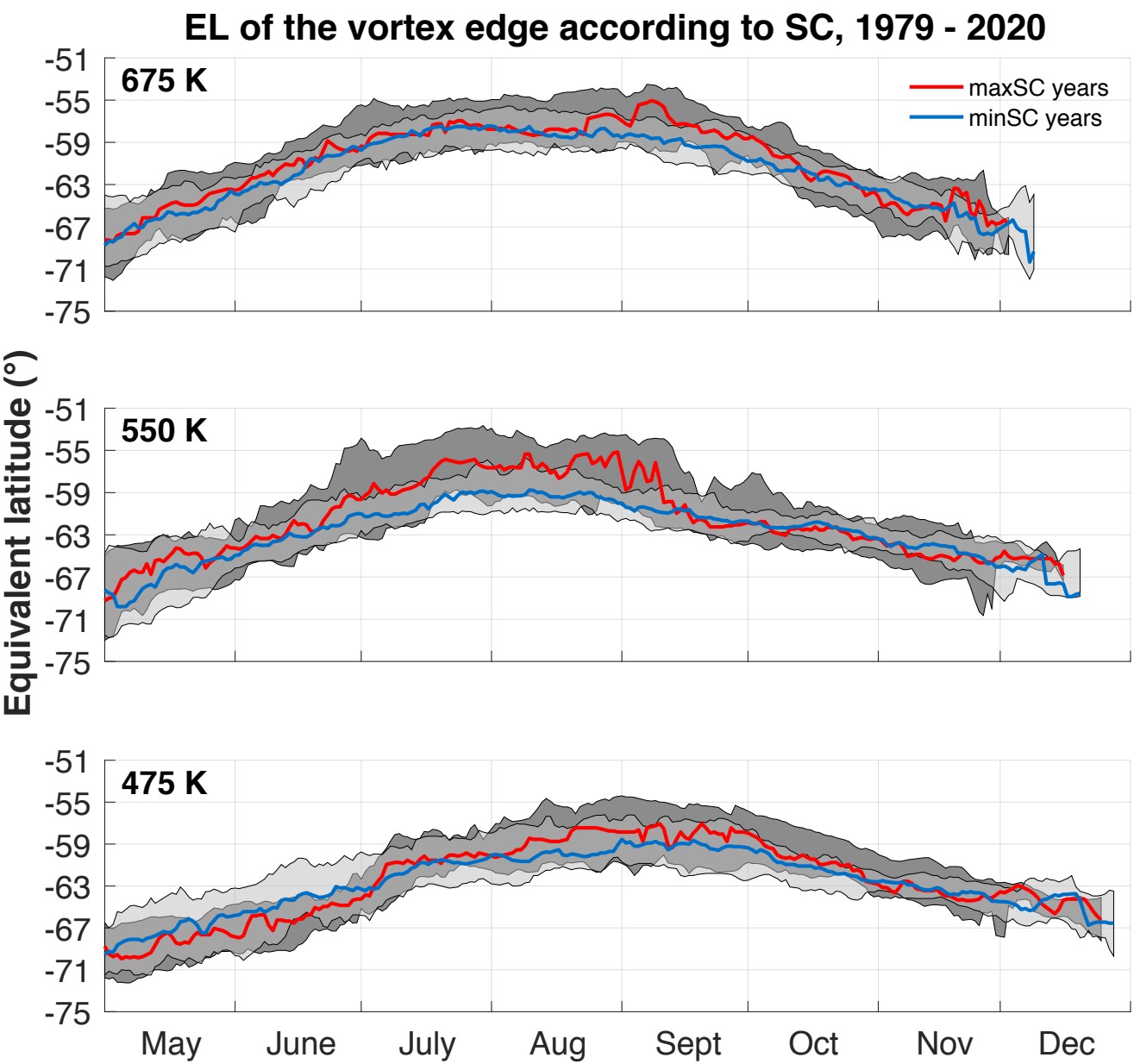

**Figure 4.** SC composites of vortex edge position's seasonal evolution according to SC for the 1979 – 2020 period, from top to bottom: 675 K, 550 K and 475 K. Red (blue) curves represent median values for maxSC (minSC) years. Dark (light) grey-filled areas indicate values between 20 and 80 percentiles for maxSC (minSC) years.

### 4.2.2 Quasi-Biennial Oscillation

We have then studied the modulation of the SC influence on the vortex edge by the QBO. Figure 5 represents the composite analysis of the polar vortex edge intensity throughout the winter for the 1979 – 2020 period at 550 K and 475 K, with maxSC and minSC years sorted with respect to the phase of the QBO: eQBO and wQBO are in the left and right panels respectively. Only results for the lower levels are shown, as the differences are less clear at 675 K. In each panel, the dark (light) grey area indicates the $20^{th}$ and $80^{th}$ percentiles of maxSC (minSC) years with the median in red (blue). Note that during the studied period there are only 5 years for maxSC/eQBO versus 10 years for minSC/eQBO, and 10 years for both maxSC/wQBO and minSC/wQBO (see Table 2).

At 550 K, both QBO phases are characterized by a stronger vortex edge during minSC years but the differences between minSC and maxSC medians are largest during eQBO years. Largest variability of vortex edge intensity for minSC years (with largest observed values) is also seen for eQBO years. During wQBO phase, minSC years show a larger duration of the period of maximum intensity (from September to November) and maxSC years are characterized by a stronger vortex edge and a longer vortex duration, compared to their equivalent during eQBO phases. A similar behavior of the vortex edge intensity is observed at 475 K. MinSC and maxSC years show respectively stronger vortex edge intensity during wQBO phase than during eQBO phase. MaxSC years are characterized by a longer vortex duration during wQBO phase than during eQBO phase. As a conclusion, the QBO further modulates the intensity of the vortex edge, especially for maxSC years, which are generally characterized by a stronger vortex edge and longer vortex duration during wQBO phase than during eQBO phase. MinSC years show also a slightly stronger vortex edge during wQBO phase.

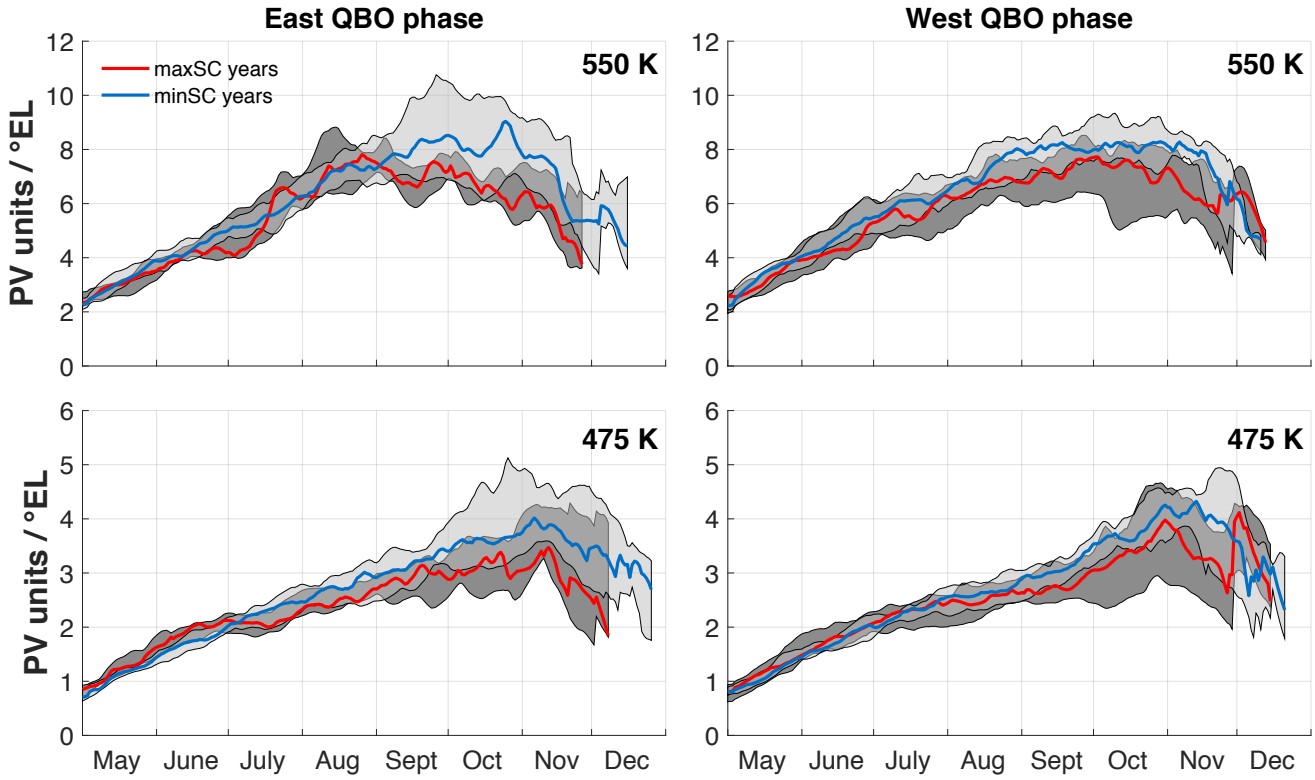

**Figure 5.** Composites of vortex edge intensity's seasonal evolution according to SC and QBO for the 1979 – 2020 period, from top to bottom: 550 K and 475 K. Left (right) panels represents eQBO (wQBO) phases. Red (blue) curves represent median values for maxSC (minSC) years. Dark (light) grey-filled areas indicate values between 20 and 80 percentiles for maxSC (minSC) years.

### 4.2.3 El Niño Southern Oscillation

We have also studied the combined modulation of the polar vortex edge by both the SC and ENSO. Figure 6 displays similar composites as in Figure 5 but selecting warm (wENSO) and cold (cENSO) ENSO phases (see section 2).

5      At both 550 K and 475 K, the largest difference between minSC and maxSC median vortex edge intensity is observed for cENSO years, with minSC years still characterized by the largest intensity. MaxSC year's vortex duration is also larger during cENSO than wENSO years. At both levels, the difference between maxSC and minSC vortex edge intensity is small and insignificant during wENSO years, while cENSO are generally characterized by stronger vortex edge for both minSC and maxSC years. The polar vortex breaks earlier during the warm phase of ENSO, and especially during the maxSC years with a breakup in November. These results are in agreement with the literature (Li et al. 2016; Domeisen et al. 2019).

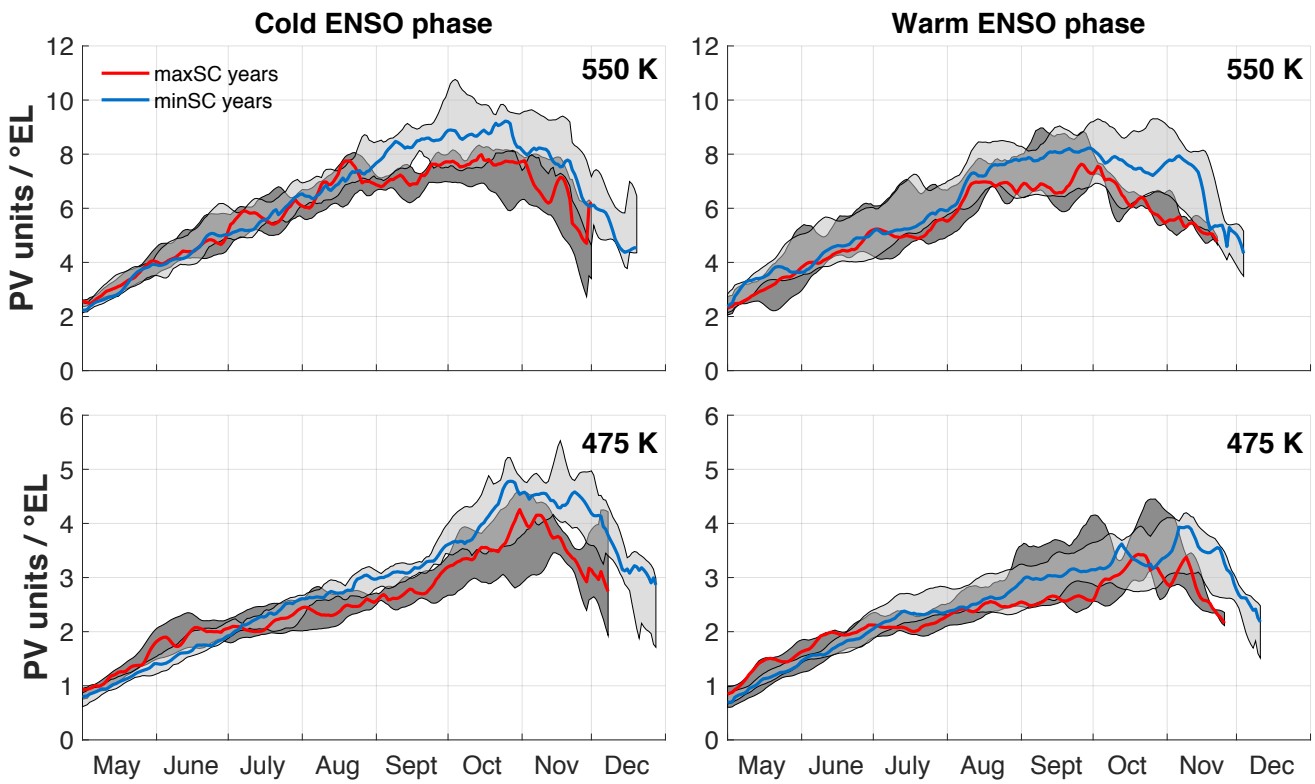

**Figure 6.** Composites of vortex edge intensity seasonal evolution according to SC and ENSO for the 1979 – 2020 period, from top to bottom: 550 K and 475 K. Left (right) panels represents cENSO (wENSO) phases (see section 2.2). Red (blue) curves represent median values for maxSC (minSC) years. Dark (light) grey-filled areas indicate values between 20 and 80 percentiles for maxSC (minSC) years.

**Table 2.** Summary of the number of years considered in the composites analyses with SC, QBO and ENSO

| Proxies | eQBO | wQBO | cENSO | wENSO |
|---------|------|------|-------|-------|
| maxSC   | 5    | 10   | 5     | 3     |
| minSC   | 10   | 10   | 7     | 5     |

## 4.3 Inter-annual evolution of the Polar Vortex edge

As seen in section 4.1, the maximum median intensity is reached during the September - November period depending on the isentropoc level. In order to study the interannual evolution of the intensity and position of the vortex edge during these periods, we identified the day when the maximum was reached at each level and averaged the parameters over ±15 days around this

5  date. Figure 7 represents the inter-annual evolution of the polar vortex edge maximum intensity at each isentropic level over the 1979 – 2020 period, averaged over September 15 – October 15, October and October 15 – November 15 at 675 K, 550 K

and 475 K, respectively. Red circles (blue squares) indicate maxSC (minSC) years. Symbol-free years are years with 10.7 cm SF values in between minSC or maxSC years.

At 550 K and 475 K, an increase of the vortex edge intensity from the beginning of the period up to the end of the 1990s is visible while this increase is not observed at 675 K. It is about 121% and 136% at 550 K and 475 K respectively between 1980 and 1996, and about 61% and 86% between 1980 and 2000 at the same levels. This increase can be attributed to the intensification of the ozone hole during the 1980s and 1990s as mentioned in other studies (Bodeker et al., 2002). From 2000, the intensity remains at a high level due to the continuing appearance of the ozone hole. Superimposed is the medium-term variability linked to the SC and inter-annual variability linked to the QBO and ENSO. In agreement with results in section 4.1 and 4.2, peaks observed around 1986, 1996, 2005 and 2016 are the signature of the 11-year solar cycle corresponding to minSC years. We note however that some maxSC years show high vortex edge intensity values, e.g. 2014 at both 550 K and 475 K levels. This year is in west QBO phase, which confirms the previous conclusion that vortex edge intensity of maxSC years is stronger during wQBO phases. However, it is in a warm ENSO phase, when the median vortex edge intensity is lower than during cENSO under maxSC conditions. It should be noted that the latest solar cycle (cycle number 24) was less intense than the previous ones (Jiang et al., 2015) and the maxSC years of the last cycle correspond to intermediate years between minimum and maximum years of the previous cycles so the modulation of the vortex edge intensity by the latest solar cycle is potentially weaker than by the earlier cycles. Similarly, while years with low edge intensity generally correspond to maxSC years, minSC years also show low intensity of the vortex edge especially at the end of the period (2016 – 2020), which corresponds to the end of the last weaker solar cycle.

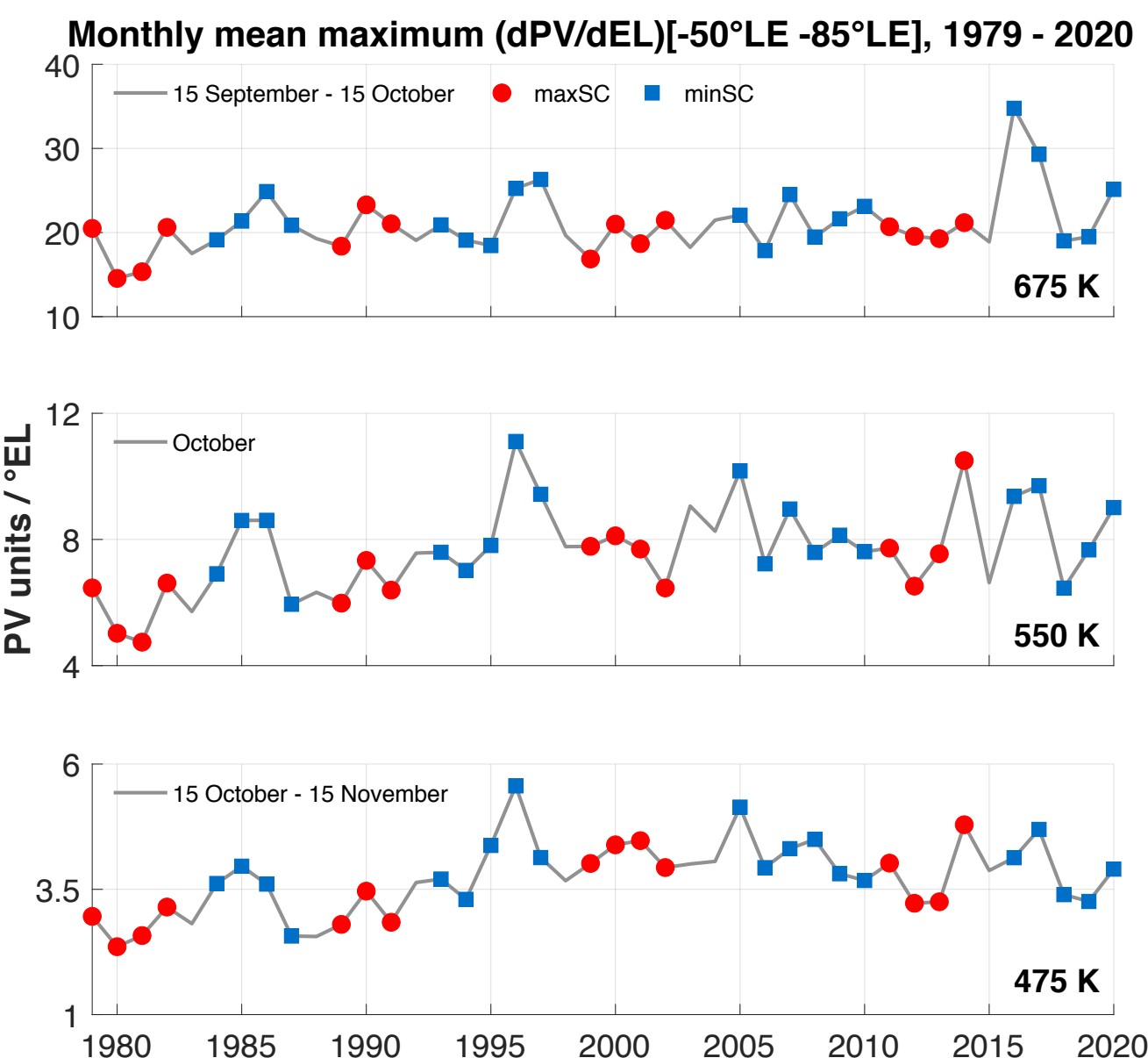

**Figure 7.** Inter-annual evolution of the maximum vortex edge intensity for the 1979 – 2020 period, averaged over September 15 – October 15 period for 675 K, October for 550 K, and the October 15 – November 15 period for 475 K. MaxSC (minSC) years are represented by red circles (blue squares).

Figure 8 represents the inter-annual evolution of the polar vortex edge position with years sorted according to the SC as described in Figure 7. The position of the vortex edge is quite similar for 550 K and 475 K levels. Between 1979 and 2001, the edge position is larger at 675 K. The most noticeable feature is the small edge position in 2002 due to the major warming and the vortex split, which occurred during that year. It was shown that the major warming in 2002, the first one observed over

Antarctica, was mainly due to increased planetary waves activities in the southern stratosphere (Hoppel et al., 2003). With the exception of this year, the maximum edge position fluctuates between -65.7°EL and -55.3°EL, at all levels. At 550 K and 475 K levels, the edge position decreases from 1981 to 1994, with values varying from -56.6°EL and -58.6°EL to -63.4°EL and -63.7°EL respectively at both levels (average decrease of 7 to 5°EL in 14 years). It can be noted that these years correspond

5 to the period when the intensity of the vortex edge increases. At 675 K, the downward trend is less visible. At all levels, particularly at 675 K and 475 K, there is a decrease in the edge position of the 2019 polar vortex, due to the minor SSW mentioned in section 4.1. Contrary to the 2002 SSW, the 2019 SSW occurred during a period of solar minimum. In contrast, the year 2020, which was characterized by a strong ozone hole with a very long duration (see section 5) does not show a particularly strong maximum vortex edge intensity value nor an outstanding value of the edge position during the respective

10 periods of maximum intensity. Later in the winter, it impacts the maximum intensity curve during a few days at the three isentropic levels.

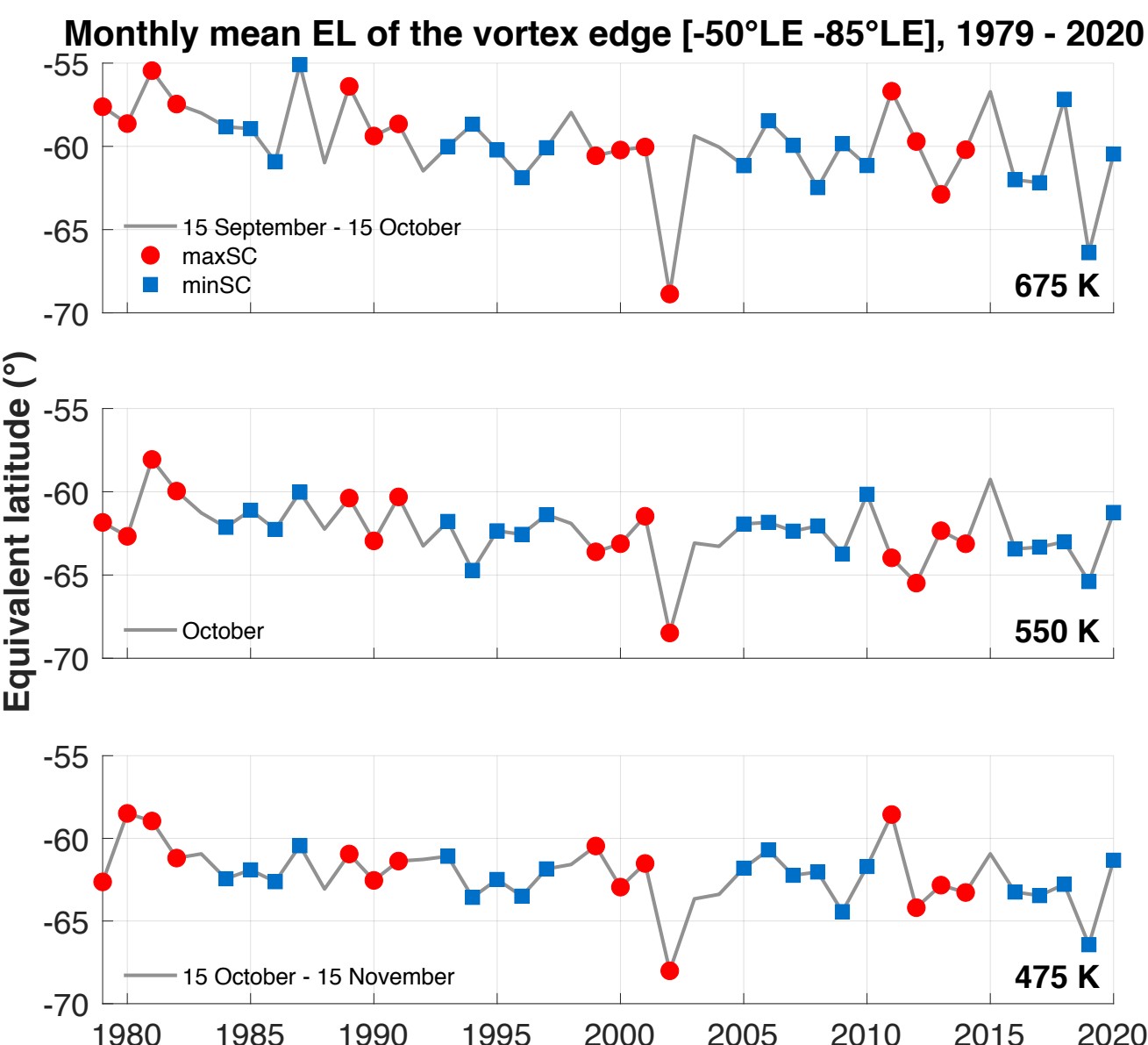

**Figure 8.** Inter-annual evolution of the vortex edge position for the 1979 – 2020 period, averaged over September 15 – October 15 period for 675 K, October for 550 K, and the October 15 – November 15 period for 475 K. MaxSC (minSC) years are represented by red circles (blue squares).

## 5 Onset and breakup of the Polar Vortex

The evolution of onset dates of the polar vortex throughout the winter from 1979 to 2020 at 675 K, 550 K and 475 K isentropic levels is displayed in Figure 9. It represents the day of the year when the polar vortex is well formed, e.g. when the horizontal

mean wind speed at the vortex edge is above the threshold values of 15.2 m.s$^{-1}$, 20 m.s$^{-1}$ and 25 m.s$^{-1}$ as suggested by Akiyoshi et al. (2009).

Due to the stronger radiative processes in the upper stratosphere, the temperature contrast between the polar region and mid-latitudes is stronger and the polar vortex forms more rapidly with a faster wind. Thus the vortex forms earlier at the highest
levels: the average day of the year the onset date occur for all thresholds combined on days 90, 98 and 108 at 675 K, 550 K and 475 K, respectively. Also, the onset date occurs later for the larger threshold values as the wind strength increases in Autumn in the polar stratosphere. The differences between onset dates according to the different threshold values decreases with altitude. At 475 K, mean values of the onset dates are days 93, 109 and 125 for the 15.2 m.s$^{-1}$, 20 m.s$^{-1}$ and 25 m.s$^{-1}$ thresholds, respectively. However, some years show large difference between the onset dates according to the different threshold values,
which can exceed one month (for example in 2002 one and a half months between 15.2 m.s$^{-1}$ and 25 m.s$^{-1}$). For example, the winter of 2002 was characterized by a difference of one and a half months between the two extreme threshold values, as the wind at the beginning of the winter was weaker compared to other winters. This is actually the first winter in which an SSW was observed, as mentioned previously. Due to the slower and less stable wind at 475 K, the vortex forms slowly and there is an important inter-annual variability of onset dates with an average difference of 32.9 days between 15.2 m.s$^{-1}$ and 25
m.s$^{-1}$ during the whole period. There are some outstanding late onset dates at 475 K, particularly for the 25 m.s$^{-1}$ threshold, e.g. on day 152 in 2002 and day 149 in 2014. In contrast the year 1992 was characterized by an early onset on day 73 for the 15.2 m.s$^{-1}$ threshold. The 550 K and 675 K levels show comparatively less variability of the onset dates for the various threshold values and the difference between the onset dates for the largest and lowest threshold values is of the order of 10 days in average (21 and 17.2 days at 550 K and 675 K respectively between the 25 m.s$^{-1}$ and 15.2 m.s$^{-1}$ threshold values).
This difference in inter-annual variability of the onset dates among the levels is further confirmed from the average standard deviation of the three thresholds curves after substracting a 3-degree polynomial. This standard deviation amounts to $\pm 8.2$ days at 475 K, which is almost two times larger than the values of the 675 K and 550 K levels ($\pm 4.8$ and $\pm 3.7$ days, respectively).

Some long-term variability in the evolution of the onset dates is also observed at the different levels. At 675 K, a decreasing trend is visible between 2010 and 2018 for the 15.2 m.s$^{-1}$ threshold, with a slightly higher inter-annual variability during
this last decade. At 550 K a similar decrease of the onset date from 2011 is observed, most pronounced for the 15.2 m.s$^{-1}$ threshold. At 475 K, the most prominent feature is a significant decline of the onset dates between 1980 and 1999 for the 25 m.s$^{-1}$ threshold value of about 29 days in 19 years, corresponding to a decline of 1.5 days per year. It is remarkable also to notice that later onset days in 2002, 2012 and 2014, correspond to years with smaller ozone holes (e.g. Pazmino et al., 2018).

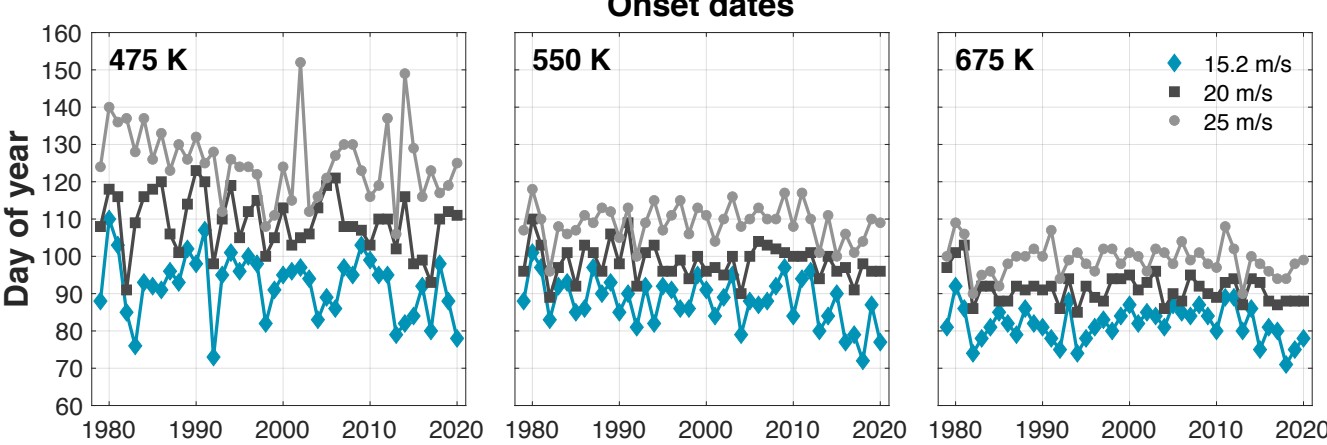

**Figure 9.** Interannual evolution of Antarctic polar vortex onset dates over the 1979 – 2020 period. Panels from left to right show onset dates at 475 K, 550 K and 675 K. Light grey, dark grey and blue curves represent onset dates for the 15.2 m.s$^{-1}$, 20 m.s$^{-1}$ and 25 m.s$^{-1}$ wind threshold values (see text).

Figure 10 shows the day when the polar vortex breaks up in Spring at 475 K, 550 K and 675 K isentropic levels. As mentioned in Nash et al. (1996), when the vortex is weakening between early and late Spring, the winds at the vortex edge also weaken, leading to the final vortex breakup. The vortex breakup is given when the horizontal mean wind speed along the vortex edge falls below the 15.2 m.s$^{-1}$, 20 m.s$^{-1}$ or 25 m.s$^{-1}$ threshold values.

The vortex forms earlier at the highest levels and it also breaks earlier: the average breakup dates for the different threshold values are days 340, 334 and 325 at 475 K, 550 K and 675 K, respectively. Rao and Garfinkel (2021) found that the average southern hemisphere stratospheric final warming at 50 hPa occurs around 2 December with JRA-55 reanalyses, which is consistent with our results at 475 K (on December 5 ). We notice some early breakup of the polar vortex: for example, in 1988 (the vortex broke up 13 days before the mean breakup date at 675 K, 20 days at 550 K and 21 days at 475 K). In 2002, the

breakup occurred 18, 9 and 8 days before the mean breakup date at 475 K, 550 K and 675 K, respectively. Some late breakups are observed during the last two decades particularly at 15.2 m.s$^{-1}$. The year 1999 is clearly noticeable at 475 K and 550 K with 21 and 27 days respectively, after the mean breakup date. The years 2007, 2008, 2010 and 2015 also stand out for the three levels: around 14 days, 15 days and 14 days after the mean breakup date at 475 K, 550 K and 675 K, respectively. Finally, the year 2020 is noticeable for its exceptionally late breakup date, with a breakup date occurring 20, 21 and 29 days after the

mean threshold dates at 475 K, 550 K, and 675 K, respectively. The value at 675 K sets a record over the whole studied period.

This figure shows that the difference between the breakup dates for the various threshold values is much smaller than for the onset dates. The average difference between breakup dates for 15.2 m.s$^{-1}$ and 25 m.s$^{-1}$ is equal to 11.5, 8.9 and 8.2 days at 475 K, 550 K and 675 K, respectively, compared to $\pm32.9$, $\pm21$ and $\pm17.2$ days, respectively, for the onset dates. The smaller differences can be explained by the important role of dynamical processes in the vortex breakup while the vortex formation

is mainly controlled by radiative processes that are less variable from one year to the next. A larger inter-annual variability

is observed for the breakup dates at the various levels and threshold values. Similarly, as for the onset dates, we calculated the standard deviation over the period after averaging the different curves of the different threshold mean and removing the long-term trend by a 3-degree polynomial. The standard deviation is equal to 10.6, 10.2 and 10.4 days at 475 K, 550 K and 675 K, respectively, compared to 8.2, 4.8 and 3.7 days for onset dates.

5    An increasing trend of the breakup dates between 1979 and 1999 is seen at all levels, which is more pronounced at 475 K. It corresponds to 35, 30 and 15 days over 21 years at 475 K, 550 K and 675 K, respectively, if we average the different threshold values at the various levels. Just after 1999 the vortex breaks up earlier. Then we observe again a later breakup of the vortex between the mid-2000s and 2010. Finally, we observe again that the vortex break up earlier, ending with the very long duration of the 2020 vortex. For all levels, a decrease in the breakup dates after 2000 is observed (apart from the extreme years like
10   2020).

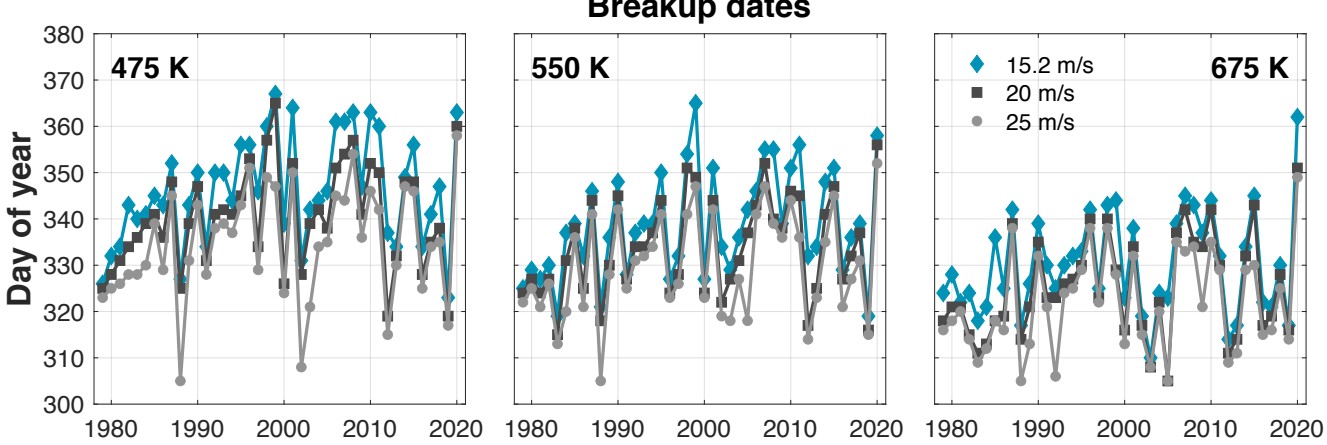

**Figure 10.** Interannual evolution of Antarctic polar vortex breakup dates over the 1979 – 2020 period. Panels from left to right show onset dates at 475 K, 550 K and 675 K. Light grey, dark grey and blue curves represent onset dates for the 15.2 m.s$^{-1}$, 20 m.s$^{-1}$ and 25 m.s$^{-1}$ wind threshold values.

**Table 3.** Summary of the onset and breakup dates

|  |  | 675 K | 550 K | 475 K |
|---|---|---|---|---|
| Onset | Average onset day over the period and for the 3 thresholds | 90 | 98 | 108 |
|  | Mean difference on the period between 25 m.s$^{-1}$ and 15.2 m.s$^{-1}$ | 17.2 | 21 | 32.9 |
|  | Std of average threshold dates after long-term trend corrected | 3.7 | 4.8 | 8.2 |
| Breakup | Average breakup day over the period and for the 3 thresholds | 325 | 334 | 340 |
|  | Mean difference on the period between 25 m.s$^{-1}$ and 15.2 m.s$^{-1}$ | 11.5 | 8.9 | 8.2 |
|  | Std of average threshold dates after long-term trend corrected | 10.4 | 10.2 | 10.6 |

## 6 Conclusion and perspectives

We have analyzed the seasonal evolution of the stratospheric polar vortex edge intensity and position in equivalent latitude in the Southern hemisphere at three isentropic levels, using ECMWF ERA-Interim data over the 1979 – 2020 period. The inter-annual evolution of the vortex edge intensity and position, as well as the onset and breakup dates at these three isentropic levels were evaluated. The studied parameters display long-term and short-term variations over the period that were analyzed using well known proxies of atmospheric variability in the stratosphere such as the solar cycle, the QBO and ENSO. Among the main results of our study, the influence of increasing ozone hole during the 1980s and 1990s on the studied parameters was clearly noticeable, confirming the results of Bodeker et al. (2002). This influence is mostly pronounced on the maximum intensity of the vortex edge, with an increase of 0.38 PV units/°EL per year at 550 K and 0.30 PV units/°EL per year at 475 K between 1980 and 1996. The vortex breakup dates show an increasing trend of 1.75 days.yr$^{-1}$, 1.5 days.yr$^{-1}$ and 0.75 days.yr$^{-1}$ at 475 K, 550 K and 675 K levels, respectively over the 1979 – 1999 period. We also find a decreasing trend over the same period for the onset dates but in this case only at 475 K and for the 25 m.s$^{-1}$ threshold value (1.5 days.yr$^{-1}$ between 1980 and 1999). We see a decreasing trend in the breakup dates after 2010 but this decrease was halted by the very long vortex duration in 2020, which set a record at the 675 K level, and also by the late breakup in 2021.

The solar cycle and to a lower extent the QBO and ENSO modulate the inter-annual evolution of the maximum intensity of the vortex edge and the breakup dates. Stronger vortex edge intensity is observed in years of solar minimum. QBO and ENSO further modulate the solar cycle influence on the vortex edge, especially at 475 K and 550 K. During wQBO phases, the difference between vortex edge intensity for minSC and maxSC years is smaller than during eQBO phases. The polar vortex edge is stronger and lasts longer for maxSC/wQBO than for maxSC/eQBO. Regarding ENSO, which has a lower impact than the QBO, the vortex edge intensity is somewhat stronger during cENSO phases for both minSC and maxSC. During this phase, the difference between minSC and maxSC medians is larger.

These results are mainly in agreement with the literature. Baldwin and Dunkerton (1998) found that the strongest influence of the QBO on the southern polar vortex occurs in late spring (November) when the final warming happens. From temperature composites at 10 hPa, they found that the vortex is slightly colder during the western phase of the QBO throughout the winter. Later, Haigh and Roscoe (2009) found that the southern stratospheric polar vortex breaks down later for combined maxSC/wQBO and minSC/eQBO years. The last two years of the study (2019 and 2020) stand out in our analysis. In 2019, the vortex maximum area was particularly small, especially at 475 K and 675 K and the vortex broke up quite early. The breakup date at 475 K and 550 K for the 15.2 m.s$^{-1}$ threshold is the lowest on record (day 323 at 475 K and 319 at 550 K). In 2020, the vortex area was not particularly large and the vortex edge not particularly strong but its duration set a record at 675 K. This very long-lasting vortex was also characterized by a strong ozone destruction (Stone et al., 2021). It will be interesting to see how the southern polar vortex evolves in the coming years.

A major perspective of our study is to extend the period analysis, using ERA5 reanalyses which are covering a longer period (from 1950) and with a higher resolution (https://www.ecmwf.int) (31 km grid for ERA5 versus 79 km for ERA-Interim). The same parameters for the more widely studied Arctic polar vortex are currently being studied for comparison between the

two hemispheres. Other factors, which particularly influence the northern hemisphere, such as the Arctic Oscillation/Northern Annular Mode, will be included in the future study.

*Data availability.* The data that support the findings of this study are openly available in [1] ECMWF ERA-Interim https://www.ecmwf.int/en/forecasts/datasets/reanalysis-datasets/era-interim [2] Solar flux at 10.7 cm ftp://ftp.seismo.nrcan.gc.ca/spaceweather/solar_flux/monthly_averages/solflux_monthly_average.txt, last access: 11 May 2021 [3] https://www.geo.fu-berlin.de/met/ag/strat/produkte/qbo/qbo.dat, last access: 11 May 2021 [4] https://www.esrl.noaa.gov/psd/enso/mei, last access: 3 January 2022. The code for the determination of the vortex edge intensity and position is available upon request to A. Lecouffe (audrey.lecouffe@latmos.ipsl.fr)

*Author contributions.* A. Lecouffe made the study and provided the results. All authors discussed the results and contributed to the final paper.

*Competing interests.* The authors declare that they have no conflict of interest.

*Acknowledgements.* The authors wish to thank Cathy Boone of Institut Pierre Simone Laplace (IPSL) for providing ERA-Interim data, and ECMWF for the availability of these data.

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
