# Peer review of "Evolution of the stratospheric polar vortex edge intensity and duration in the Southern hemisphere over the 1979 – 2020 period"

_Atmospheric Chemistry and Physics, 2021_

## Author Response (AR1)

**Answers to reviewer 1: (in green in the "tracking" version)**

We would like to thank reviewer 1 for the time taken to suggest new references and for the advices. We have generally taken these suggestions into account and modified the text. You will find below our answers to the major and minor comments.

**Major comments**

**Lack of a sufficient review on the most recent publications.**

*The impact of ENSO, solar cycle, and QBO on the polar vortex in both hemispheres has been widely and exhaustively studied in literature. However, this manuscript fails to provide a sufficient review on the most recent publications. The impact of the canonical ENSO on the SH polar vortex is insignificant in both observations and modeling studies (Rao and Ren 2020 https://doi.org/10.1007/s00382-019-05111-6, Hurwitz et al. 2011 https://journals.ametsoc.org/view/journals/atsc/68/4/2011jas3606.1.xml).*

*The impact of QBO on the stratosphere is also reported in the latest literature (Rao et al. 2020 https://doi.org/10.1175/JCLI-D-19-0663.1; Butchart et al. 2019 https://doi.org/10.5194/gmd-11-1009-2018).*

*As the solar cycle's impact on the SH stratospheric polar vortex, it is also discussed most recently in Figure 3 of Rao et al. 2020JGR (https://doi.org/10.1029/2020JD03272). I suggest the authors to explore more of recent publications to see what has been done and what has not.*
* * *
Thanks for the many advices on literature. We have added some of the references suggested in the introduction for the general presentation of the use of these factors in the literature:

Page 3 lines 23-28: "Domeisen et al. (2019) (https://doi.org/10.1029/2018RG000596) have indicated that the El Niño events are associated with a warming and weakening of the polar vortex in the polar stratosphere in both hemispheres, and Li et al. (2016) have shown that early breakup of the southern polar vortex occurs during El Niño events. In contrast, Rao and Ren., (2020) did not find a significant impact of the canonical ENSO index on the Southern Hemisphere polar vortex in both observations and modeling studies. With indices of Niño-3 and Niño-4 regions, Hurwitz et al., (2011) (https://doi.org/10.1175/2011JAS3606.1) have shown that during "warm pool event" (positive SST in Niño-4 regions) the heat flux is higher and the Antarctic vortex breaks up earlier."

Regarding the impact of ENSO, we would first like to remind that our study is focused on the intensity and position of the polar vortex edge, parameters that are generally not considered in other studies, which makes comparisons sometimes difficult. Following both reviewers remarks on the use of ENSO index, we have revised our analysis in section 4.2.3 and have excluded years with MEI.v2 values between the - 0.5 and + 0.5 as recommended by NOAA [4] (https://psl.noaa.gov/enso/mei/). With the revised calculation of the ENSO MEI.v2 index, we found, by removing the neutral years, that the polar vortex breaks up early during the warm ENSO years phase. See the new Figure 6 below with the combined SC and ENSO indices, plus the composite analysis with ENSO only.

Please see that references like « [4] » are defined at the end of the documents, in the « data availability » section.

We did not include a sensitivity study based on the use of Niño 3 and Niño 4 index as in Hurwitz et al., (2011) but we think it is a good idea for future work.

At the end of section 2.2 we have changed the sentence as follows:

Page 5 lines 20-22: "Then mean ENSO over the period is sorted to distinguish La Niña, characterized by negative values below -0.5 MEI.v2 (cold ENSO), and El Niño by positive values higher than +0.5 MEI.v2 (warm ENSO). Then 10 wENSO and 14 cENSO years are considered in this study."

The new Figure 6 with the combined SC and ENSO indices is as follows:

[Figure]

Figure 6: Composites of vortex edge intensity seasonal evolution according to SC and ENSO for the 1979 – 2020 period, from top to bottom: 550 K and 475 K. Left (right) panels represents cENSO (wENSO) phases (see section 2.2). Red (blue) curves represent median values for maxSC (minSC) years. Dark (light) grey-filled areas indicate values between 20 and 80 percentiles for maxSC (minSC) years.

The following figure (not shown in the article), displays the composite analysis with ENSO only. This figure is requested and described in the minor comment section 4.2.2, 4.2.3.

[Figure]

Figure: ENSO composites of vortex edge intensity's annual seasonal evolution for the 1979 – 2020 period, from top to bottom: 675 K, 550 K and 475 K. Red (blue) curves represent median values for wENSO (cENSO) years. Dark (light) grey-filled areas indicate values between 20 and 80 percentiles for wENSO (cENSO) years.

**Discussion on the 2002 SSW but lack of the 2019 SSW**

*Ample evidence has reported the similarity of the 2002 and 2019 SSWs in the SH. The paper discusses the main characteristics of the polar vortex edge in 2002 but fails to mention the 2019 SSW. Related studies are also ignored in the paper. The main characteristics of the SH polar vortex during the 2019 SSW have also been reported in Rao et al. 2020JGR ([https://doi.org/10.1029/2020JD032723](https://doi.org/10.1029/2020JD032723)), Shen et al. 2020GRL ([https://doi.org/10.1029/2020GL089343](https://doi.org/10.1029/2020GL089343)). The background of this study is still lacking and the references can be further improved.*
* * *
The 2019 sudden stratospheric warming has indeed been mentioned in the original version of the article (see page 16, lines 1-2) but we agree that this event deserves more attention in the article. See below new information and references added at various locations in the manuscript about the 2019 SSW.

Page 9, lines 4-7: "The 2019 winter impacts the minimum curve during the last 2 weeks of September at 675 K and is located between the minimum curve and the 20th percentile from September until the beginning of November for each level. During this year, a minor SSW

occurred at the end of August, which displaced and weakened the polar vortex. The stratospheric polar vortex abruptly weakened and warmed on August 25th (Lim et al., 2021) (https://doi.org/10.1175/BAMS-D-20-0112.1). MERRA2 analyses showed a rapid 50 K increase of polar temperature at 10 hPa between September 5 and September 11 (Yamasaki et al., 2020) (https://doi.org/10.5194/acp-20-5111-2020). Minimum values of winds at 10 hPa and 60°S were found on September 18 (Rao et al., 2020) (https://doi.org/10.1029/2020JD032723). This event induced the smallest Antarctic ozone hole on record. Although it appeared earlier in August, the ozone hole reached an area of 15 million km² by September 1, but decreased to an area of 8 million km² by September 17 (Lim et al., 2021) (https://doi.org/10.1175/BAMS-D-20-0112.1)."

This figure (not shown in the article), supports the paragraph above, and shows the 2019 winter in purple, plotted among other winters, superimposed on the curves of Figure 2 of the article:

[Figure]

Page 20 line 5-7: "At all levels, particularly at 675 K and 475 K, there is a decrease in the edge position of the 2019 polar vortex, due to the minor SSW mentioned in section 4.1. Contrary to the 2002 SSW, the 2019 SSW occurred during a period of solar minimum"

**Start date of the SH polar vortex and the final warming date**

*In my understanding, the start date of the stratospheric polar vortex in both hemispheres are mainly forced radiatively due to the annual cycle, and it should be very stable. I checked the start date of the polar vortex in the NH using the zonal mean zonal wind at 60N and 10hPa as the threshold. I found that the start date of the NH polar vortex is very stable. In contrast, the*

*final warming date in both hemispheres differs from year to year due to the dynamics associated with planetary wave activities. However, this study shows that both the start date and end date of the SH polar vortex have a large interannual variability. What forces such a strong variation of the start date of the stratospheric polar vortex. The authors also failed to mention the most recent studies on the final warming date in the SH (Rao and Garfinkel 2021CD, https://doi.org/10.1007/s00382-021-05647-6). They also discussed the possible impact of the ozone depletion and recovery on the polar vortex final warming date.*
* * *
In our study, a larger inter-annual variability is observed for the breakup dates than for the onset dates at the various levels and threshold values. We have calculated the standard deviation over the period after averaging the curves of the different thresholds and removing the long-term trend by a 3-degree polynomial. The standard deviation for the onset dates are equal to 8.2, 4.8 and 3.7 days at 475 K, 550 K and 675 K, compared to 10.6, 10.2 and 10.4 days, respectively.

For the onset dates, the variability is indeed more important for the 475 K level.
At this level, the wind is slower and less stable. It results in a slowly vortex formation with an important inter-annual variability of onset dates.

We did not add the reference to Rao and Garfinkel 2021CD (https://doi.org/10.1007/s00382-021-05647-6), because in this study the break up date is determined at 10 hPa level which is too high for the comparison with our results.

**Minor comments**

**P1L18: This sentence should be supported by some citations. Please insert.**

We have added the Randel and Newman, (1998) (https://doi.org/10.1007/978-1-935704-10-2_9) reference as follows:

Page 1 lines 18-19: "It appears due to the seasonal cooling associated with the decrease of solar radiation above the pole (Randel and Newman, 1998)."

**L24: The most recent report by Rao and Garfinkel 2021CD checked the interannual variation of the final warming date from CMIP5/6 models and JRA55 reanalysis. Explore if you missed more recent reports.**

The sentence is just very generic. We have added some references as well as the suggested one, as follows:

Page 1 lines 24-25, page 2 line 1: "Over Antarctica, the polar vortex is generally present from April until December with a large variability in the breakup dates resulting from the year-to-year variability of dynamical processes in the stratosphere (Waugh and Randel 1999 (https://doi.org/10.1175/1520-0469(1999)056<1594:COAAAP>2.0.CO;2); Rao and Garfinkel 2021)."

**P2L1: The reference put too much on the ozone depletion, but review on other aspects of the stratospheric polar vortex is insufficient.**

This part of the introduction was changed as follows in order to reduce the reference to ozone depletion and insert new statement about the importance of the southern polar vortex.

We removed the sentence "PSCs are found to be much more abundant in the Antarctic polar vortex as compared to the Arctic polar vortex due to the increased stability of the southern polar vortex. As a result of the asymmetry in polar vortices intensity, the southern vortex experiences much colder temperatures in winter, which results in stronger ozone depletion over a large area (the so-called ozone hole)." on page 2 lines 11-15 of the original document

We added a paragraph about the stratospheric vortex impact on the surface as follows:

Page 2 lines 21-27: "The polar vortex also has an impact on the climate surface in both hemispheres. Indeed, studies have shown an effect of the stratospheric polar vortex displacements on cold spells in the northern hemisphere, in North America (Tripathi et al., 2015) (https://doi.org/10.1002/qj.2432). In the southern hemisphere, others have shown that a weak vortices can have an influence on the surface climate in Australia. Lim et al. (2019) (https://doi.org/10.1038/s41561-019-0456-x) have highlighted that selected years of lower vortex intensity results in higher temperatures and less precipitation over eastern Australia. The dramatic weakening of the Antarctic vortex in 2019 had a large impact on meteorological conditions over the country that resulted in the strong Australian fires of the turn of the year 2019/2020.

**L13: The ozone depletion events are also existing in the NH. The AUG organized one special issue for the NH ozone loss event in the 2019/2020 winter:**

**https://agupubs.onlinelibrary.wiley.com/doi/toc/10.1002/(ISSN)1944-8007.ARCTICSPV. Choose several references and discuss the ozone depletion in the NH (e.g., Garfinkel 2020, 2021; Feng et al. 2021). The authors really should read more to enrich the introduction of the paper. This version is really not satisfactory.**

Our paragraph does not exclude the fact that there is no ozone loss in the Arctic. In fact, the authors are well aware of ozone depletion in the Arctic and have contributed to the evaluation of Arctic ozone depletion with several articles. We have however added more references and inserted a new sentence as follows:

Page 2 lines 4-6: "Ozone loss occurs in both hemispheres. This loss is variable in the northern hemisphere as many studies have shown (Solomon et al., 1999 (https://doi.org/10.1029/1999RG900008); Goutail et al., 2005 (https://doi.org/10.5194/acp-5-665-2005); Pommereau et al., 2018 (https://doi.org/10.1016/j.crte.2018.07.009); WMO 2018 (http://ozone.unep.org/science/assessment/sap); Grooß and Müller, 2020 (https://doi.org/10.1029/2020JD033339))."

**L17-19: The future recovery of the ozone and its possible impact on the vortex in both hemispheres are also discussed in Rao and Garfinkel 2021CD.**

We have inserted the following references in the sentence:

Page 2 lines 20-21: "Many studies document this phenomenon (e.g. WMO 2018 and references therein)."

**L28: What is wind module? Please specify.**

The following equation defines de wind module W:

$$W = \sqrt{U^2 + V^2}$$

Where U is the meridional wind and V is the zonal wind.

After correction and suggestions made by the reviewer 2, "wind module" has been replaced by "wind mean speed" throughout the text.

**P3L2-3: The impact of QBO, ENSO, and solar cycle on the polar vortex in both hemispheres have been widely studied in literature. Please be more exhaustive when you mention the most recent studies. Please see my major comments.**

Done by adding citations and suggestions in the text as described in our answer to the reviewer's first major comment.

**L9-10: This conclusion is also reported by Rao et al. 2020JGR when they checked the possible impact of the QBO phase on the 2019 SSW in SH.**

In this paragraph, we refer to studies that consider several winters. We thus added the following sentence and reference:

Page 3 line 19-20: "Camp and Tung (2007) (https://doi.org/10.1029/2006GL028521) supports this finding that the state of the northern hemisphere polar stratosphere is less perturbed during solar cycle minimum and westerly QBO phases."

**L12-19: This part should be moved to the method section. Or I suggest to remove or shorten.**

We understand this part is not clear. Here is the mentioned paragraph:

"Several methods have been suggested in order to determine the onset and breakup dates of the polar vortex. They are based on a minimum area computed from equivalent latitudes (Manney et al. 1994 (https://doi.org/10.1029/94GL02368); Zhou et al. 2000 (https://doi.org/10.1029/1999GL011018)) or wind speed thresholds at the edge of the vortex (e.g. Nash et al. 1996 (https://doi.org/10.1029/96JD00066)). Manney et al. (1994) and Zhou et al. (2000) consider that the vortex breaks down and disappears when its size falls below 1% of the Earth's surface, or when its edge position is larger than 78.5°EL. More recently, Millan et al. (2020) (https://doi.org/10.5194/acp-2020-1181) compared the polar vortex evolution with different reanalyses, including ERA-Interim. Results showed that all reanalyses where in agreement with the reanalysis ensemble mean (REM), which shows that we can be confident with the ERA-Interim reanalyses for our study."

Here is the new paragraph:

Page 3 lines 28-30: "Several methods have been suggested in order to determine the onset and breakup dates of the polar vortices. They are based on a minimum vortex area computed from equivalent latitudes (Manney et al. 1994; Zhou et al. 2000) or wind speed thresholds at the edge of the vortex (e.g. Nash et al. 1996). The latter is used in WMO (2018) to calculate the dates at which the Arctic and Antarctic polar vortex breaks each spring."

And we have moved the underlined part above, to the section 2.1:

Page 4 line 15-18: "Recently, Millan et al. (2020) compared the polar vortex evolution with different reanalyses, including ERA-Interim. Results showed that all reanalyses where in agreement with the reanalysis ensemble mean (REM), which shows that we can be confident with the ERA-Interim reanalyses for our study."

**L24: You might emphasize the novelty of this study, because the possible impact of QBO, ENSO, and solar cycle have been widely reported.**

We have added this sentence at the end of the paragraph.

Page 4, lines 1-2: "This is the first study of the variability of the Antarctic stratospheric polar vortex edge and persistence over a long period (42 years)."

**P4L7-12: This sentence is toooo.... long. Can you split this sentence and clearly describe the model MIMOSA. Please tell readers what MIMOSA consist of and how it can predict the PV. Is it a forecast model?**

MIMOSA can be used as a forecast model.

The sentence mentioned is: "The MIMOSA model is a three-dimensional high-resolution PV advection model (Hauchecorne et al., 2002) (https://doi.org/10.1029/2001JD000491) which has been used to analyze, among other studies, the permeability of the southern polar vortex to volcanic aerosols from Cerro Hudson and Mount Pinatubo eruptions in 1991 (Godin et al., 2001) (https://doi.org/10.1029/2000JD900459), to predict the 10 extension in the lower mid-latitude stratosphere of polar and subtropical air masses (Heese et al., 2001) (https://doi.org/10.1029/2000JD900818), or to evaluate average total ozone evolution within the Antarctic vortex with PV fields simulated by the model, used to determine the vortex position in Pazmino et al. (2018) (https://doi.org/10.5194/acp-18-7557-2018). »

The Reviewer 2 suggest to reword or remove the end of the sentence but we propose to modify the entire section 2.1 paragraph on pages 4 as follows:

"PV fields are calculated from ECMWF ERA-Interim reanalysis [1] (Dee et al., 2011) (https://doi.org/10.1002/qj.828). As these reanalyses end in August 2019, we used the operational data from ECMWF from September 2019 until December 2020. Recently, Millan et al. (2020) compared the polar vortex evolution with different reanalyses, including ERA-Interim. Results showed that all reanalyses where in agreement with the reanalysis ensemble mean (REM), which shows that we can be confident with the ERA-Interim reanalyses for our study. ERA-Interim temperature, geopotential and wind data with a resolution of 1.125° latitude x 1.125° longitude are inputs to the MIMOSA model, which is a three-dimensional high-resolution PV advection model (Hauchecorne et al., 2002). From MIMOSA high resolution PV fields, it is possible to follow the evolution of polar air masses and filamentation processes of the polar vortex. Sampled every 6 hours, ERA-Interim reanalyses are interpolated on selected isentropic surfaces. The model computes PV and EL fields on isentropic surfaces with a resolution of 0.3° latitude x 0.3° longitude, using a polar projection centered on the South from 90°S to 10°N. The advection method is applied to this orthographic grid. After some time, the MIMOSA grid is distorted by the horizontal gradients of the wind fields. A re-interpolation of the PV fields on the original grid every 6 hours is then performed. Finally, in order to take into account diabatic processes, a relaxation of the MIMOSA advected PV (APV) towards the ECMWF PV is made every 12 hours with a 10 day time constant. This model has been used to analyze, among other studies, the permeability of the southern polar vortex to volcanic aerosols from Cerro Hudson and Mount Pinatubo eruptions in 1991 (Godin et al., 2001), and to predict the extension in the lower mid-latitude stratosphere of polar and subtropical air masses (Heese et al., 2001). In Pazmino et al. (2018), PV fields simulated by the model are used to evaluate average total ozone evolution within the Antarctic vortex. For this study, PV fields are computed at 675 K, 550 K and 475 K isentropic levels."

**L18: Rao et al. 2019 JGR also used this index to select the solar max and min years. Please refer to Table 3 in Rao et al. 2019 JGR (https://doi.org/10.1029/2019JD030826)**

We did not cite any reference here because a lot of other studies use this index.

**Section 2.2: L20: 23th ⇒ 21st**

The sentence has been rewritten differently as reviewer 2 suggested:

Page 5, line 8: "Data were obtained for solar cycles 21 to 24 (1976 to 2020)."

**Section 2.2 : L25: If you read Rao et al. 2019 JGR (supplementary material), please mention that they also consider the intensity change for each solar cycle.**

We do not find the « supplementary material » section here https://agupubs.onlinelibrary.wiley.com/doi/full/10.1029/2019JD030826 but a « support information »

On page 5 lines 8-11, we added: "Years characterized by minimum and maximum solar intensity were selected from the difference of maximum and minimum intensity of each cycle (a methodology also considered in Rao et al., 2019). The minimum (maximum) intensity threshold was defined as the lower (upper) third of this difference, so that the minimum and maximum thresholds are different for each cycle.

**L27-30: This index is not reasonable, because it can not distinguish between Eastern and Central Pacific ENSO events. Only CP ENSO can impact the SH polar vortex. Previous studies (Hurwitz et al. 2011; Rao and Ren 2020) have reported that EP ENSO is not related with the SH stratosphere. Please change to use Nino3 and Nino4 index and revisit the possible impact of ENSO on the SH polar vortex edge.**

We addressed this comment in our answer to the reviewer's major comment #1

We have changed the figure description of the section 4.2.3 for El Niño Southern Oscillation as follows, and have added a sentence:

Page 16 lines 8-9: "The polar vortex breaks earlier during the warm phase of ENSO, and especially during the maxSC years with a breakup in November. These results are in agreement with the literature (Li et al. 2016 (https://doi.org/10.1175/JCLI-D-15-0816.1); Domeisen et al. 2019).

Table 2 was changed as follows:

| Proxies | eQBO | wQBO | cENSO | wENSO |
|---------|------|------|-------|-------|
| maxSC | 5 | 10 | 8 5 | 7 3 |
| minSC | 10 | 10 | 11 7 | 9 5 |

**L31: This sentence can be removed. It has been mentioned earlier.**

The sentence has been removed.

Page 5 lines 6-8 have been modified as follows: "For our study, we averaged the 10.7 cm solar flux and other proxies over the May - November period, which corresponds to the time period when the Southern polar vortex is formed."

**L33: This classification of ENSO state is also weird. 21 warm ENSO and 21 cold ENSO. Why is there no neutral ENSO state? Rather weird and unacceptable.**

Please see the answer to the previous comment on page 5 lines 20-22.

We have changed the sentence as follows:

Page 5 lines 20-22: "Then mean ENSO over the period is sorted to distinguish La Niña, characterized by negative values smaller than -0.5 MEI.v2 (cold ENSO), and El Niño by positive values higher than +0.5 MEI.v2 (warm ENSO). Then 10 wENSO and 14 cENSO years are considered in this study."

**P5 Table 1: ENSO index should be changed.**

As indicated previously, we did not change the index, only the calculation method.

**L5: The position of the edge is in the unit of EL, rather than as a function of EL degree. Please correct throughout the paper. The authors might misunderstand the function. The edge is a single value, independent of the EL. Edge = edge(theta, time). But PV = PV(EL, theta, time). Mathematically, the description is incorrect.**

The sentence referred to is page 6 lines 3-5 on the original document: "the method described in Nash et al. (1996) is used, which consists in determining the position of the edge from the maximum PV gradient weighted by the wind mean speed as a function of EL."

We find that "as a function of EL" is correct here because it addresses how the vortex edge is determined (from the maximum of grad(PV/EL) x wind(EL)).

However, we agree that throughout the text as in description of figures (e.g. figures 2 and 3), the wording is not correct and have changed "as a function of equivalent latitude" to "in equivalent latitude".

**L11-12: PV is not an output for the NCEP/NCAR reanalysis. I do not think the authors clearly know and understand what they read. If the PV is also obtained from MIMOSA driven by the NCEP/NCAR reanalysis, please clarify.**

Again, the authors understand and can read scientific articles. In the Manney et al., 1994c (https://doi.org/10.1029/94GL02368) study, the authors do not use MIMOSA driven by the NCEP/NCAR reanalysis. Manney et al., 1994c use data from the US national Meteorological Center and PV on isentropic surfaces calculated from NMC data describe in their previous article (Manney et al., 1994b) (https://doi.org/10.1175/1520-0469(1994)051<2973:OTMOAT>2.0.CO;2) and explain as follows: "Sixteen years of geopotential height and temperature data from the US National Meteorological Center (NMC) [Finger et al. 1993 and references there in] are used. Rossby-Ertel potential vorticity (PV) on isentropic surfaces calculated from NMC data [e.g., Manney et al. 1994b] is used to describe the evolution of the lower stratospheric vortex."

So in Manney et al., (1994c) they use PV **computed from** the NCEP/NCAR reanalyses, previously known as "US National Meteorological Center (NMC)". We have clarified the sentence as follows:

Page 6, lines 10-12: "Manney et al. (1994) first determined that the breakup date corresponds to the date when the EL of a chosen PV contour at the 465K level is greater than 80°, using PV data **computed from** the National Centers for Environmental Prediction and the National Center for Atmospheric Research (NCEP/NCAR) reanalyses."

**P6L6: I did not see any special value of using so many thresholds. The results are different for those thresholds. Which one should readers believe?**

The sentence is "In this study, we use the Nash et al. (1996) method to determine the vortex onset and breakup dates, used also in WMO (2019), with three threshold values (15 m.s$^{-1}$, 20 m.s$^{-1}$and 25 m.s$^{-1}$) following Akiyoshi et al. (2009)."

Akiyoshi et al., (2000) (https://doi.org/10.1029/2007JD009261) do not specify clearly why they use the horizontal wind speeds of 20 and 25 m.s$^{-1}$ added to the one use by Nash et al., (1996). But they compared Nash et al., (1996) threshold value method with the Langematz and Kunze (2006) (https://doi.org/10.1007/s00382-006-0156-2) method which consist in defining the breakup date as the day of the year when the zonal mean westerlies at 65°S and 50 hPa decrease below a threshold value of 10 m.s$^{-1}$. Our objective of using different thresholds is thus to evaluate the sensitivity of the onset and breakup dates to the selected threshold values.

We have changed the sentence to:

Page 6, lines 26-28: "In this study, we use the Nash et al. (1996) method to determine the vortex onset and breakup dates, also used in WMO (2018). Two threshold values (20 m.s$^{-1}$ and 25 m.s$^{-1}$) following Akiyoshi et al. (2009) are added to this method, in order to evaluate the sensitivity of the onset and breakup dates to the chosen threshold values (see section 5)."

**L11: See above. What is function?**

Page 7 line 5: We changed with "in EL".

**P7L6: Add discussion for the 2019 SSW in SH. Please inserted relevant references.**

This comment has been addressed in our answer to the reviewer's major comment # 2.

**Figure 2: See above. Function is misleading.**

After the suggestion of reviewer 2, we have changed the legend of Figure 2 as follows:

Page 10 legend: "Evolution of daily position of the vortex edge in equivalent latitude as a function of time over the 1979 - 2020 period, from top to bottom: 675 K, 550 K and 475 K. Median values are represented by the black bold curve. Blue filled areas show values between 20 and 80 percentiles, while thin black curves represent the maximum and minimum values over the period."

**P8L4: Those are factors which might control the interannual variation of the polar vortex. This sentence should be rephrased.**

The sentence mentioned is "Proxies such as the solar cycle, QBO and ENSO are used to describe the polar vortex edge interannual variation over the 1979 - 2020 period."

We have changed to:

Page 11 lines 2-3: "Factors such as the solar cycle, QBO and ENSO are used to describe the interannual variability in the temporal evolution of the polar vortex edge over the 1979 - 2020 period."

**P9L10: How did you test the difference for the medians? The difference for the mean can be tested using the t-test. Which test is used for the medians? Please clarify.**

Indeed, the purpose of the t-test is to observe the difference between means. For the medians, I therefore used the Mann-Whitney test.

We have changed the end of the paragraph from:

"The t-test results indicate that differences are significant from mid-September to the end of October with a mean p-value of 0.023 at 675K, from September to late November with p value of 0.032 at 550 K, and during the same period with a p-value of 0.023 at 475 K." page 9 line 10 of the original document.

to:

Page 11 lines 13-16: "A Mann-Whitney test was performed to characterize the significance of these differences. The Mann-Whitney test results indicate that differences are significant from 27 September to 26 October at 675 K, from 9 to 24 September and from 3 October to 21 November at 550 K, and from 19 September to 15 October and from 11 to 26 November at 475 K."

**P10L1: delete "statistical"**

We have deleted "statistical" and have changed to "composite analysis".

**L8: How did you test the difference for the medians?**

As for the previous comment, we have modified the test method and have used the Mann-Whitney test.

We have changed the end of the paragraph from:

"Differences between the medians are largest during the July - August period at 550K with a mean p-value of 0.03" page 10 line 8 of the original document.

to:

Page 13 lines 5-7: "The difference between the medians was assessed by a Mann-Whitney and differences are significant from 9 to 18 September at 675 K, from 18 July to 11 August and from 27 August to 7 September at 550K, and from 15 to 20 June at 475 K."

**Section 4.2.2, 4.2.3: The two subsections still focused on the impact of solar cycle on the polar vortex edge. I prefer to seeing the results for the composite for ENSO and QBO directly. Can you also show?**

The two following figures are the composite analysis for QBO and ENSO only, represented as for the SC in Figure 3.

The composite analysis of QBO alone does not highlight any different behavior between the QBO phases so we did not include it in the article.

[Figure]

Figure: QBO composites of vortex edge intensity's annual seasonal evolution for the 1979 – 2020 period, from top to bottom: 675 K, 550 K and 475 K. Red (blue) curves represent median values for wQBO (eQBO) years. Dark (light) grey-filled areas indicate values between 20 and 80 percentiles for wQBO (eQBO) years.

ENSO alone shows us that the vortex edge is stronger during the cold ENSO phase, especially at the 550 K and 475 K levels. Earlier breakups are also observed during the warm ENSO phase. We have not included this figure in the text because Figure 6 shows very well that we have a stronger vortex edge during the cold ENSO phase and an early break up during warm ENSO.

[Figure]

Figure: ENSO composites of vortex edge intensity's annual seasonal evolution for the 1979 – 2020 period, from top to bottom: 675 K, 550 K and 475 K. Red (blue) curves represent median values for wENSO (cENSO) years. Dark (light) grey-filled areas indicate values between 20 and 80 percentiles for wENSO (cENSO) years.

**P14L5: In my understanding, the maximum day is different from year to year. But you fixed from 15 September to 15 October. If so, remove this sentence as is misleads readers.**

We have fixed periods of maximum intensity from 15 September to 15 October at 675 K, in October at 550 K and from 15 October to 15 November at 475 K.

We do agree what the paragraph between lines 1 and 7 on page 14 of the original version is not clear. The following paragraph:

"As seen in section 4.1, the maximum median intensity is reached from September to late October at 675K, from September to early November at 550K, and early November at 475K. In order to study the interannual evolution of the maximum intensity and position of the vortex edge during these periods, we identified the day when the maximum was reached at each level and averaged the parameters over ±15 days around this date. Figure 7 represents the inter-annual evolution of the polar vortex edge maximum intensity at each isentropic level over the 1979 - 2020 period, averaged over September 15 – October 15, October and October 15 – November 15 at 675K, 550K and 475K respectively."

has been changed to:

Page 17, lines 2-6 and page 18 line 1: "As seen in section 4.1, the maximum median intensity is reached during the September – November period depending on the isentropic level. In order to study the interannual evolution of the intensity and position of the vortex edge during these periods, we identified the day when the maximum was reached at each level and averaged the parameters over ±15 5 days around this date. Figure 7 represents the inter-annual evolution of the polar vortex edge maximum intensity at each isentropic level over the 1979 - 2020 period, averaged over September 15 – October 15, October and October 15 – November 15 at 675K, 550K and 475K respectively".

**L9-11: The discussion is useless. Those so-called decrease and increase reflect the interannual variation, rather than any trend.**

We do not agree with this comment. We observe a visible trend at 550 K and 475 K between 1980 and 2000 in Figure 7, as quantified in our manuscript. Bodeker et al., (2002) (https://doi.org/10.1029/2001GL014206) documents well the intensification of the dynamical containment of Antarctic ozone depletion during the 1980s and 1990s, which can be related to what we observe.

**L19-20: The final solar cycle is weak, mention in Rao et al. 2019 JGR.**

We have added the Jiang et al., 2015 reference as this study is based on the last 11 years cycle of solar activity less vigorous than the previous three cycles.

The sentence with the reference added is as follows:

Page 18 lines 13-16: "It should be noted that the latest solar cycle (cycle number 24) was less intense than the previous ones (Jiang et al., 2015) (https://doi.org/10.1088/2041-8205/808/1/L28) and the maxSC years of the last cycle correspond to intermediate years between minimum and maximum years of the previous cycles so the modulation of the vortex edge intensity by the latest solar cycle is potentially weaker than by the earlier cycles."

**P15L3: Figure ⇒ figure**

The word has been changed.

**P16L2: Rao et al. 2020JGR also reported the 2019 SSW in SH.**

This sentence refers to the link between the ozone hole and the 2019 SSW. The reference Rao et al., (2020) in JGR does not refer to the ozone hole, only to the 2019 SSW.

**P18L1: You show the final warming date. In my understanding, this is another method of determining the final warming date. The mean date is consistent with Rao and Garfinkel 2021CD. You might mention the reference to support your results.**

The mentioned sentence is: "Figure 10 shows the day when the polar vortex breaks up in Spring due to the final vortex warming, at 475K, 550K and 675K isentropic levels."

We have removed the part "due to the final vortex warming," of the sentence:

Page 23 lines 1-2: Figure 10 shows the day when the polar vortex breaks up in Spring at 475K, 550K and 675K isentropic levels."

**L16: Figure ⇒ figure**

The word has been changed.

**P20L2: as a function of ⇒ in the units of**

Changed to "in equivalent latitude".

**P21L5: ⇒ in the future study.**

Done.

**Answers to reviewer 2: (in violet in the "tracking" version)**

We would like to thank the reviewer for taking the time to correct mistakes in our article and for suggesting improvement in the writing of the manuscript. We have taken these suggestions into account and modified the text. Here are our answers to the reviewer's major and minor comments.

**Major comments**

**Section 2.1:** **Why is the MIMOSA model needed to determine PV fields. It can be determined directly from the ECMWF data. Does MIMOSA provide higher time and spatial sampling?**

We use the MIMOSA model to obtain higher spatial resolution PV fields. The MIMOSA model allows better tracking of polar air masses (the continuity of vortex-related structures, such as filaments for example).

For example, in the first figure below, the PV fields of ERA 5 are used directly, at 475 K with a resolution of 0.25° on 5 December 1997 at 00UT.

In the second figure, the ERA5 fields are used on the same date, but as inputs to the MIMOSA model. The noisy filament seen running from 60°N/60°E to 40°N/10°W across Europe on the second figure is better represented by MIMOSA fields. We can also see some dubious high PV values in the ERA5 figure around 40°N/45°E that are not visible in the MIMOSA figure.

[Figure]

[Figure]

Reviewer 1 asked for more information about the model for the readers, and to give better explanations on how it works. So we have changed the entire paragraph as follows in section 2.1 pages 4 and 5:

"PV fields are calculated from ECMWF ERA-Interim reanalysis [1] (Dee et al., 2011) (https://doi.org/10.1002/qj.828). As these reanalyses end in August 2019, we used the operational data from ECMWF from September 2019 until December 2020. Recently, Millan et al. (2020) (https://doi.org/10.5194/acp-2020-1181) compared the polar vortex evolution with different reanalyses, including ERA-Interim. Results showed that all reanalyses where in agreement with the reanalysis ensemble mean (REM), which shows that we can be confident with the ERA-Interim reanalyses for our study. ERA-Interim temperature, geopotential and wind data with a resolution of 1.125° latitude x 1.125° longitude are inputs to the MIMOSA model, which is a three-dimensional high-resolution PV advection model (Hauchecorne et al., 2002) (https://doi.org/10.1029/2001JD000491). From MIMOSA high resolution PV fields it is possible to follow the evolution of polar air masses and filamentation processes of the polar vortex. Sampled every 6 hours, ERA-Interim reanalyses are interpolated on selected isentropic surfaces. The model computes PV and EL fields on isentropic surfaces with a resolution of 0.3° latitude x 0.3° longitude, using a polar projection centered on the South from 90°S to 10°N. The advection method is applied to this orthographic grid. After some time, the MIMOSA grid is distorted by the horizontal gradients of the wind fields. A re-interpolation of the PV fields on the original grid every 6 hours is then performed. Finally, in order to take into account diabatic processes, a relaxation of the MIMOSA advected PV (APV) towards the ECMWF PV is made every 12 hours with a 10 day time constant. This model has been used to analyze, among other studies, the permeability of the southern polar vortex to volcanic aerosols from Cerro Hudson and Mount Pinatubo eruptions in 1991 (Godin et al., 2001) (https://doi.org/10.1029/2000JD900459), and to predict the extension in the

lower mid-latitude stratosphere of polar and subtropical air masses (Heese et al., 2001) (https://doi.org/10.1029/2000JD900818). In Pazmino et al. (2018) (https://doi.org/10.5194/acp-18-7557-2018), PV fields simulated by the model are used to evaluate average total ozone evolution within the Antarctic vortex. For this study, PV fields are computed at 675 K, 550 K and 475 K isentropic surfaces."

**Section 2.2: For the definitions of years with QBOe and QBOw as well as SCmin and SCmax, the upper and lower third of the distribution, is used, but why is this not done for ENSO (no intermediate years). Why is the one-third limit for each solar cycle determined separately?**

The lower and upper third of the distribution is used for the Solar Cycle only, not for the QBO. For the QBO, the east and west phase are determined with positive values (wQBO) and negative values (eQBO). For ENSO, as recommended by NOAA (https://psl.noaa.gov/enso/mei/) and thanks to reviewer 1's comments, the MEI.v2 index for ENSO has been calculated with MEI.v2 values between the - 0.5 and + 0.5 MEI.v2 representing the intermediate (neutral) years. At the end of section 2.2 we have changed the sentence as follows:

Page 5 lines 20-22: "Then mean ENSO over the period is sorted to distinguish La Niña, characterized by negative values smaller than -0.5 MEI.v2 (cold ENSO), and El Niño by positive values higher than +0.5 MEI.v2 (warm ENSO). Then 10 wENSO and 14 cENSO years are considered in this study."

The one third limit for each solar cycle is determined separately because of the variability of the different cycles (for example the 25th cycle does not reach the same maximum values as the 22th, 23th and 24th ones).

*__p6,l2:__ "other studies", please cite them here (see also introduction)*

Page 6, section 3.2, lines 23-24, we added some citations which refers to the ozone depletion on the polar vortex lifetime.

"The authors joined other studies (Atkinson et al. 1989 (https://doi.org/10.1038/340290a0); Müller et al. 2008 (https://doi.org/10.5194/acp-8-251-2008); Zhang et al. 2017 (https://doi.org/10.1155/2017/3078079)) in concluding that the vortex lifetime is influenced by the ozone depletion during spring."

*__Conclusion:__ Are the trends in the breakup dates after 2000 (the tendency for earlier breakup dates) a new result? Should it not be then highlighted in the abstract? Also, "trends are not halted" just by adding one year (2020) which has been extreme.*

To our opinion, the trends in the breakup dates after 2000 are the sign of decadal variability rather than a long-term trend. So we have not added this result in the abstract.

About the 2020 late breakup, we wrote in the initial manuscript "A seemingly decreasing trend in the breakup dates after 2010 was halted by the very long vortex duration in 2020, which set a record at the 675K level."

We have changed the sentence to:

On page 25 lines 12-14: "We see a decreasing trend in the breakup dates after 2010 but this decrease was halted by the very long vortex duration in 2020, which set a record at the 675 K level, and also by the late breakup in 2021."

We thank very much the reviewer for the time taken to suggest accurate corrections and questions in our manuscript file. We have implemented the suggested changes in spelling and grammar suggested by the reviewer. We will not list each small point in this document. The answers to questions in the pdf file are listed here.

**P1.L5: A bit hard to understand, you mean the vortex area has the lowest day-to-day(?) variability. Or could this be due to larger uncertainties in the vortex parameters before and after this period.**

The sentence refers to the variability of the position of the vortex edge in equivalent latitude. The vortex edge position has indeed the lowest year to year variability during the period of maximum intensity as shown in Figures 1 & 2. Since the southern polar vortex is well formed before and after this period, this feature is not due to larger uncertainties during these periods.

In order to clarify the statement, the following paragraph:

"The vortex edge is stronger in late winter, over September - October - November with the period of strongest intensity occurring later at the lowermost level. A lower variability of the edge position is observed during the same period."

Has been changed to:

Page 1 lines 4-6: "The vortex edge is stronger in late winter, over September - October - November with the period of strongest intensity occurring later at the lowermost level. At the same period, we observe a lower variability of the edge position."

**P2.L28: "" -> "wind mean speed"**

We have changed "wind module" throughout the paper to "wind mean speed".

**P2.L28: This is correct but not a good explanation, suggest to rephrase by describing: EL defines the latitude limit of the (polar cap) area which exceeds a certain PV value (Maximum PV is then given at EL=90degs). A PV field sorted by EL will then make the polar vortex concentric around the pole**

Thanks for the suggestion, which better explains the notion of equivalent latitude. we have included in our manuscript. The changed part of the paragraph is as follows:

Page 2 lines 34-35 and page 3 lines 1-3: "Numerous studies on the vortex boundary definition have been performed. Nash et al. (1996) (https://doi.org/10.1029/96JD00066) defined the vortex edge as the location of the maximum PV gradient as a function of equivalent latitude (EL), weighted by the wind module. EL defines the latitude limit of the polar area which exceeds a certain PV value (maximum PV is then given at EL = 90 degrees e.g. Butchart and Remsberg (1986) (https://doi.org/10.1175/1520-0469(1986)043<1319:TAOTSP>2.0.CO;2)). A PV field sorted by EL will then make the polar vortex concentric around the pole. This is the method used in this study."

**P4.L11/12: somehow disconnected to the sentence, please reword (or just ommit)**

We do agree this sentence is too long on page 4 lines 7-12 of the original document:

"The MIMOSA model is a three-dimensional high-resolution PV advection model (Hauchecorne et al., 2002) which has been used to analyze, among other studies, the permeability of the southern polar vortex to volcanic aerosols from Cerro Hudson and Mount Pinatubo eruptions in 1991 (Godin et al., 2001), to predict the extension in the lower mid-latitude stratosphere of polar and subtropical air masses (Heese et al., 2001), or to evaluate average total ozone evolution within the Antarctic vortex with PV fields simulated by the model, used to determine the vortex position in Pazmino et al. (2018). For this study, PV fields are computed at 675K, 550K and 475K isentropic surfaces." Page 4 lines 7-12 of the original document.

This sentence has been changed to:

Page 4 lines 26-30: "This model has been used to analyze, among other studies, the permeability of the southern polar vortex to volcanic aerosols from Cerro Hudson and Mount Pinatubo eruptions in 1991 (Godin et al., 2001), and to predict the extension in the lower mid-latitude stratosphere of polar and subtropical air masses (Heese et al., 2001). In Pazmino et al. (2018), PV fields simulated by the model are used to evaluate average total ozone evolution within the Antarctic vortex."

**P4.L30: not clear what is said here. Bi.-monthly averages reflect the time-delayed response to ENSO. What is meant with "seasonality of ENSO"?**

We do agree this is not a good explanation. This was reformulated as follows:

Page 5 lines 18-20: "Referring to the NOAA description of the MEI.v2 index (see data availability [4]): "The EOF are calculated for 12 overlapping bi-monthly "seasons" in order to take into account ENSO's seasonality, and reduce effects of higher frequency intra-seasonal variability.""

Where the reference [4] correspond of the NOAA website in the paper in « data availability » https://www.esrl.noaa.gov/psd/enso/mei

**P4.L33: why not define only upper and lower third of the distribution as cENSO and wENSO (like QBO and solar flux).**

Please refer to the answer to the second major comment above.

**P5.L5:  → "horizontal wind"**

After the reviewer's suggestion in page 2 line 28 of the original document, we have changed "wind module" throughout the paper to "wind mean speed".

**P5.L7: W is the wind speed**

Please see the answer to the comment above.

**P6.L2: references**

Please see the answer to the reviewer's third major comment above.

**P6.L12: not clear to me. you mean that 80% of all years fall within the the upper and lower limit (why 20%).**

The blue filled area indicates daily values between 20 and 80 percentiles over the whole period (1979 – 2020). The upper curve of the blue area refers to the 80 percentiles and the lower curve of the blue area refers to the 20 percentiles. We mean that the complete blue area represents 80-20 = 60 inter-percentiles of the total values.

**P7.legend: show  80 percentiles**

This suggestion was made throughout the text. We do not agree with that because values in blue represents values between the 20th and 80th percentiles.

**P7.L3: maximum vortex edge position". sounds awkward, wouldn't it be better to say "maximum vortex area"**

This is not a vortex area but the position of the edge at the maximum PV gradient forced by the wind mean speed. We do agree this is not the maximum vortex edge position, but only the vortex edge position.

We have changed the sentence by removing the word "maximum" now in page page 8 line 2.

**P7.L5 : cannot be both, here minima.**

We agree that it is a mistake. We have removed the word "maxima" now in page page 8 line 5.

**P8.legend: ""**

We have removed the word "maximum".

**P8.legend: ""**

As suggested, we have replaced "as a function of equivalent latitude" with "in equivalent latitude as a function of time".

**P9.L5: "represents  80th percentiles"**

We do not agree with that suggestion. Please see answers to comments on page 6 line 12 and page 7 legend, in the original version.

**P9.L6: "strong vortex edge" "vortex edge intensity". The terms sound to me a bit strange. The higher the PV gradient (wind speed) at the vortex edge, the more stable is the polar vortex. Not the vortex edge is stronger, but the entire vortex is stronger and more stable.**

In this study, we focus on the edge of the polar vortex and did not consider other parameters such as the mean PV value within the boundary of the vortex, which could be the scope of another study with other parameters such as the mean temperature within the vortex.

**P10.legend: "" -> "seasonal"**

We have changed « annual » to « seasonal » throughout the text.

**P11.titre figure: " EL of the vortex edge according to SC, 1979 – 2020"**

We have removed the word "maximum" in the titles of the Figures 2 and 4.

**P14.L6-7: These periods do not agree with the periods from the first sentence in this paragraph.**

As reviewer 1 also noted it was not clear, we have changed the following paragraph from:

"As seen in section 4.1, the maximum median intensity is reached from September to late October at 675K, from September to early November at 550K, and early November at 475K. In order to study the interannual evolution of the maximum intensity and position of the vortex edge during these periods, we identified the day when the maximum was reached at each level and averaged the parameters over ±15 5 days around this date. Figure 7 represents the inter-annual evolution of the polar vortex edge maximum intensity at each isentropic level over the 1979 - 2020 period, averaged over September 15 – October 15, October and October 15 – November 15 at 675K, 550K and 475K respectively."

to:

Page 17, lines 2-6 and page 18 line 1: "As seen in section 4.1, the maximum median intensity is reached during the September – November period depending on the level. In order to study the interannual evolution of the maximum intensity and position of the vortex edge during these periods, we identified the day when the maximum was reached at each level and averaged the parameters over ±15 5 days around this date. Figure 7 represents the inter-annual evolution of the polar vortex edge maximum intensity at each isentropic level over the 1979 - 2020 period, averaged over September 15 – October 15, October and October 15 – November 15 at 675 K, 550 K and 475 K respectively".

**P15.L1: "Figure 8 represents the inter-annual evolution of the  polar vortex edge position with years sorted according to the SC as described in Figure 7"**

We have removed the word "maximum".

**P16.L4: it may exceed normal values later than the period considered in Fig. 8.**

Thanks for this relevant comment. Please see the figure below, with some special winters plotted, including the 2019 winter in violet.

We have modified the sentence to:

Page 20, lines 7-11: "In contrast, the year 2020, which was characterized by a strong ozone hole with a very long duration (see section 5) does not show a particularly strong maximum vortex edge intensity value nor an outstanding value of the edge position during the respective periods of maximum intensity. Later in the winter, it impacts the maximum intensity curve during a few days at the three isentropic levels."

[Figure]

**P17.L3: why is this expected? Explain**

The original sentence is: "As expected, the vortex forms earlier at the highest levels: the average day of the year the onset date occurs for all thresholds combined is on days 90, 98 and 108 at 675K, 550K and 475K respectively."

We have completed the sentence to answer to this question:

Page 22 lines 3-6: "Due to the stronger radiative processes in the upper stratosphere, the temperature contrast between the polar region and mid-latitudes is stronger and the polar vortex forms more rapidly with a faster wind. Thus the vortex forms earlier at the highest levels: the average day of the year the onset date occur for all thresholds combined is on days 90, 98 and 108 at 675 K, 550 K and 475 K, respectively."

**P17.L9: not logical, the major warming occurred after the onset. A bit more explanation is needed here.**

The sentence mentioned is: "This year was however characterized by the first major warming observed in Antarctica, as mentioned previously."

We have changed this sentence and have added more explanation as follows:

Page 22 lines 10-13: "For example, the winter of 2002 was characterized by a difference of one and a half months between the two extreme threshold values, as the wind at the beginning of

the winter was weaker compared to other winters. This is actually the first winter in which an SSW was observed, as mentioned previously."

**P17.L10: why is it important?**

The sentence mentioned is: "Also, the inter-annual variability of onset dates is rather important at 475 K."

We have added more explanation as follows:

Page 22 lines 13-15: "Due to the slower and less stable wind at 475 K, the vortex forms slowly and there is an important inter-annual variability of onset dates with an average difference of 32.9 days between 15.2 m.s$^{-1}$ and 25 m.s$^{-1}$ during the whole period."

**P17.L1: which level?**

This is at 475 K.
We have changed "at this level" to "at 475 K" in page 22 line 15.

**P18.L14: express this in days after the mean breakup date.**

The mentioned sentence is: "Finally, the year 2020 is noticeable for its exceptionally late breakup date, with an average breakup for the three thresholds on days 360, 355 and 354 at 475K, 550K, and 675K, respectively."

It has been changed to:

Page 23, lines 13-15: "Finally, the year 2020 is noticeable for its exceptionally late breakup date, with a breakup date occurring 20, 21 and 29 days after the mean threshold dates at 475 K, 550 K, and 675 K, respectively."

**P18.L17-18: Are these standard deviations?**

These are not standard deviation but the average difference between onset dates for the two extreme threshold values (15.2 m.s$^{-1}$ and 25 m.s$^{-1}$) at 475 K, 550 K and 675 K compared to the breakup dates line 17. The standard deviation for the mean threshold values are 10.6, 10.2 and 10.4 days for 475 K, 550 K and 675 K for breakup dates (see page 24 lines 3-4).

**P19.L7: It seems to me that the trend is rather negative (shorter duration) after 1999 apart from the very extreme years as stated in the next sentence.**

The mentioned sentence is "The vortex is thus persisting later after 1999." and is about Figure 10.

We have changed the sentence to:

Page 24 lines 7-9: "Just after 1999 the vortex breaks up earlier. Then we observe a later breakup of the vortex between the mid-2000s and 2010. After 2010, we observe again that the vortex breaks up earlier, ending with the very long duration of the 2020 vortex."

**P20.L15: A trend is not halted just because of one extreme year at the end of the period.**

The mentioned sentence is "A seemingly decreasing trend in the breakup dates after 2010 was halted by the very long vortex duration in 2020, which set a record at the 675K level."

We have reworded the sentence as follows:

Page 25 lines 12-14: "We see a decreasing trend in the breakup dates after 2010 but this decrease was halted by the very long vortex duration in 2020, which set a record at the 675 K level, and also by the late breakup in 2021."

**P20.L17-19: sentence too long, split it.**

The mentioned sentence is "Stronger vortex edge intensity is observed in years of solar minimum. QBO and ENSO further modulate the solar cycle influence on the vortex edge, especially at 475K and 550K: during wQBO phases, the difference between vortex edge intensity for minSC and maxSC years is smaller than during eQBO phases."

We have separated the sentences as follows:

Page 25 lines 16-18: "Stronger vortex edge intensity is observed in years of solar minimum. QBO and ENSO further modulate the solar cycle influence on the vortex edge, especially at 475 K and 550 K. During wQBO phases the difference between vortex edge intensity for minSC and maxSC years is smaller than during eQBO phases."

**P20.L21 : make a separate sentence out of it**

The mentioned sentence is "Regarding ENSO, which has a lower impact than the QBO, the vortex edge intensity is somewhat stronger during cENSO phases for both minSC and maxSC, and the difference between minSC and maxSC medians is larger."

We have separated the sentences as follows:

Page 25 lines 19-21: "Regarding ENSO, which has a lower impact than the QBO, the vortex edge intensity is somewhat stronger during cENSO phases for both minSC and maxSC. During this phase, the difference between minSC and maxSC medians is larger."

**P20.L31: better to cite papers on 20202 Antarctic ozone depletion from the JGR/GRL special issue.**

We have removed both links (https://public.wmo.int/en/media/news/2020-antarctic-ozone-hole-large-and-deep and https://public.wmo.int/en/media/news/record-breaking-2020-ozone-hole-closes) and supported the sentence with a new citation from (Stone et al., 2021) (https://doi.org/10.1029/2021GL095232).

The new sentence is:

"This very long-lasting vortex was also characterized by a strong ozone destruction (Stone et al., 2021). It will be interesting to see how the southern polar vortex evolves in the coming years."